# Functionally asymmetric motor neurons contribute to coordinating locomotion of *Caenorhabditis elegans*

Oleg Tolstenkov[1,2,3], Petrus Van der Auwera[1,2,4], Wagner Steuer Costa[1,2], Olga Bazhanova[1], Tim M Gemeinhardt[1,2], Amelie CF Bergs[1,2,5], Alexander Gottschalk[1,2,3]*

[1]Buchmann Institute for Molecular Life Sciences, Goethe University, Frankfurt, Germany; [2]Institute for Biophysical Chemistry, Goethe University, Frankfurt, Germany; [3]Cluster of Excellence Frankfurt Macromolecular Complexes, Goethe University, Frankfurt, Germany; [4]Department of Biology, Functional Genomics and Proteomics Unit, Katholieke Universiteit Leuven, Leuven, Belgium; [5]International Max Planck Research School in Structure and Function of Biological Membranes, Frankfurt, Germany

**Abstract** Locomotion circuits developed in simple animals, and circuit motifs further evolved in higher animals. To understand locomotion circuit motifs, they must be characterized in many models. The nematode *Caenorhabditis elegans* possesses one of the best-studied circuits for undulatory movement. Yet, for 1/6[th] of the cholinergic motor neurons (MNs), the AS MNs, functional information is unavailable. Ventral nerve cord (VNC) MNs coordinate undulations, in small circuits of complementary neurons innervating opposing muscles. AS MNs differ, as they innervate muscles and other MNs asymmetrically, without complementary partners. We characterized AS MNs by optogenetic, behavioral and imaging analyses. They generate asymmetric muscle activation, enabling navigation, and contribute to coordination of dorso-ventral undulation as well as anterio-posterior bending wave propagation. AS MN activity correlated with forward and backward locomotion, and they functionally connect to premotor interneurons (PINs) for both locomotion regimes. Electrical feedback from AS MNs via gap junctions may affect only backward PINs.

DOI: https://doi.org/10.7554/eLife.34997.001

*For correspondence:
a.gottschalk@em.uni-frankfurt.de

Competing interests: The authors declare that no competing interests exist.

## Introduction

Locomotion represents a basic component of many complex behaviors and is regulated by neuronal circuits that share similar properties in a wide variety of species, including humans (*Guertin, 2012*; *Kiehn, 2011*; *Mullins et al., 2011*). These circuits, as shown in virtually all model systems studied, can generate rhythmic motor patterns without sensory inputs, and therefore act as central pattern generators (CPGs; *Pearson, 1993*). In higher vertebrates CPGs are very complex systems and represent distributed networks made of multiple coupled oscillatory centers, grouping in pools as discrete operational units (*Fidelin et al., 2015*; *Kiehn et al., 2010*; *Rybak et al., 2015*). Motor neurons (MNs) within the pool are able to integrate convergent inputs; they are recruited and activated gradually, which underlies the variable changes in muscle tension that are necessary for movement. Overlaid on these circuits are interactions with (and among) premotor interneurons (PINs), which modulate the patterns of MN activity and coordinate the CPGs (*Goulding, 2009*). In mammals, it is particularly the commissural interneurons (CIN) which regulate activity of left and right CPGs, and which may therefore themselves act as rhythm generators. Such neurons are often excitatory (e.g.

CINei of the mouse; **Kiehn, 2016**), but can also be inhibitory (e.g. CINi), while others act as electrical connectors or activity 'sinks' for CPGs and motor neuron pools. Examples are ipsilateral V2a interneurons in zebrafish, which are retrogradely recruited by motor neurons (**Song et al., 2016**) to modulate their activity. A similar effect of motor neurons providing feedback to the CPG was recently shown for drug-induced locomotor-like activity in the neonatal mouse (**Falgairolle et al., 2017**).

Despite the difference in the forms of locomotion and anatomy of neural circuits between vertebrates and invertebrates, they share similar principles. Yet, how complex vertebrate locomotion circuits operate and how they developed from more simple ones is not understood, thus a comprehensive analysis of invertebrate circuits is a prerequisite to this goal. The relative simplicity of invertebrate nervous systems has helped to develop concepts that guide our understanding of how complex neuronal networks operate (**Marder et al., 2005**; **Selverston, 2010**). *C. elegans* is a nematode with only 302 neurons in the hermaphrodite. A largely reconstructed wiring diagram of its neural circuits (**Varshney et al., 2011**; **White et al., 1986**) and various tools for imaging and (opto) genetic interrogation of circuit activity (**Fang-Yen et al., 2015**; **Leifer et al., 2011**; **Nagel et al., 2005**; **Stirman et al., 2011**) render *C. elegans* a useful model to study fundamental principles of the neuronal control of behavior.

*C. elegans* moves by generating waves of dorso-ventral bends along its body. These predominantly lead to forward movement, which is occasionally interrupted by brief backing episodes, the frequency of which is modulated by sensory responses (**Cohen and Sanders, 2014**; **Gjorgjieva et al., 2014**; **Pierce-Shimomura et al., 2008**; **Zhen and Samuel, 2015**). The animal's undulations are controlled by neural circuits in the head and VNC. The core components of the motor circuits in *C. elegans* include head motor/interneurons that exhibit $Ca^{2+}$ oscillations during alternating head bending (**Hendricks et al., 2012**; **Shen et al., 2016**). The bending motions may be transmitted to the remainder of the body in part by proprioceptive feedback (**Wen et al., 2012**), with contribution also by gap junctions (**Xu et al., 2018**). Furthermore, rhythm generators in the VNC were shown to play a role in oscillatory activity during locomotion of *C. elegans* (**Fouad et al., 2018**). In the body, motor neurons are found in ensembles or subcircuits, repeating 6 times from the 'neck' to the tail of the animal, containing one or two neurons of each class (6 – 13 neurons found in the individual classes, with 11 AS MNs; **Haspel and O'Donovan, 2011**; **White et al., 1986**). Upstream of the motor neurons are PINs which integrate inputs from sensory and other interneurons, and that relay their activity in a gating fashion: They are themselves not oscillatory, but set up- or down-states of the motor neurons, using gap junction and synaptic networks (**Kawano et al., 2011**), in a manner similar to the V2a interneurons of the fish (**Song et al., 2016**). The classes of MNs are distinguished by transmitter used (acetylcholine or GABA), ventral or dorsal innervation, and roles in forward or backward locomotion (**Von Stetina et al., 2005**; **Zhen and Samuel, 2015**). Functions of the different types of MNs are understood to various degrees. For example, the DA9 A-type MN was recently demonstrated to generate intrinsic rhythmic activity by P/Q/N-type $Ca^{2+}$ channels, which is both attenuated and potentiated by activity of the reversal PIN AVA (**Kawano et al., 2011**; **Gao et al., 2018**). Thus, motor neurons, rather than interneurons, can be oscillators, demonstrating that different activities are compressed in the *C. elegans* motor circuit with its limited number of cells. To fully understand these circuits, all of the motor neurons need to be characterized. However, for the cholinergic AS MN class, representing one fifth of VNC cholinergic neurons, surprisingly no physiological data is available. Yet, these neurons are interesting in that they asymmetrically innervate only dorsal muscle and ventral inhibitory VD neurons. Further, in contrast to other MN types, the AS MNs are innervated extensively by chemical synapses from both forward and reverse PINs, and they also form gap junctions with these cells (**White et al., 1986**).

In this study, we investigated the role of AS MNs in the VNC locomotor circuit based on predictions made from the wiring diagram, using optogenetic tools, behavioral analysis, and $Ca^{2+}$ imaging in immobilized and moving animals. We reveal important roles of AS MNs in dorso-ventral and antero-posterior coordination of undulations during locomotion, as stimulation of AS MNs distorts, and inhibition blocks, propagation of the body wave. We show that AS MNs act through excitation of dorsal muscles and inhibitory ventral VD motor neurons. The intrinsically evoked activity of AS MNs during crawling correlates with both forward and reverse locomotion. Functionally asymmetric electrical connections suggest AS MN feedback to the backward PIN AVA, a feature recently observed for locomotor circuits also in other animals (**Matsunaga et al., 2017**; **Song et al., 2016**).

## Results

### Selective expression and activation of optogenetic tools in AS MNs

Six classes of cholinergic MNs are involved in mediating the dorso-ventral sinusoidal wave observed during locomotion of *C. elegans*: DA, VA, DB, VB, VC and AS. Up to date, no promotor exclusively triggering expression in AS MNs is known. To achieve specific activation of AS MNs, we used a subtractive approach for expression combined with selective illumination. The p*unc-17* promoter (*unc-17* encodes the vesicular acetylcholine transporter) drives expression in all cholinergic neurons including the MNs in the VNC. In combination with p*acr-5* (expression in DB and VB MNs) and p*unc-4* (expression in DA, VA and VC MNs), we could restrict expression of optogenetic tools to the AS MNs (*Figure 1A,B*): Briefly, broad expression from p*unc-17* was suppressed in the DB, VB, DA, VA and VC neurons by expressing dsRNA constructs targeting the optogenetic tool using p*acr-5* and p*unc-4* promoters (*Figure 1AI*). Alternatively, we used the Q system (*Wei et al., 2012*): We placed the QF transcriptional activator under the p*unc-17* promoter, thus driving expression of the optogenetic tool from constructs harboring the QUAS QF binding motif. To restrict expression to AS MNs in the VNC, we additionally used the QS suppressor under the p*acr-5* and p*unc-4* promoters (*Figure 1AII*). We verified that expression along the VNC was restricted to AS MNs, by imaging many animals (*Figure 1B*; *Figure 1—figure supplement 1A*). Last, since these approaches still led to expression in additional cholinergic neurons in head and tail ganglia, we avoided activation of optogenetic tools in those cells by selective illumination of segments of the animals body that correspond to AS MNs (*Stirman et al., 2011*; *Stirman et al., 2012*; *Figure 1AIII*).

### Depolarization of AS MNs activates body wall muscles (BWMs) and increases body bending during locomotion

*C. elegans* moves by propagating undulation waves along the body. Body bends are generated by cholinergic neurons mediating contraction of muscles on one side, and by GABAergic neurons mediating simultaneous relaxation of the contralateral side of the body (*Donnelly et al., 2013*; *McIntire et al., 1993*). According to the wiring diagram (*Chen et al., 2006*; *Varshney et al., 2011*; *White et al., 1986*) AS MNs send 68/144 documented synapses to the dorsal BWM cells (i.e. 47% of all presynaptic contacts made by the AS neurons) and 66/144 synapses to inhibitory ventral (GABAergic) VD MNs (i.e. 46% of all AS neuron synaptic terminals). We measured parameters of crawling in intact worms moving freely on agar substrate. Activation of ChR2 in all cholinergic neurons including VNC MNs leads to strong contraction of the worm body and coiling (*Zhang et al., 2007*; *Liewald et al., 2008*). AS MNs innervate only dorsal muscles, thus we wondered if their simultaneous depolarization would hinder propagation of the body bending wave. Animals in which ChR2 was activated in AS MNs kept the ability to propagate the undulation, yet they displayed a distorted wave, deeper bending (*Figure 1C,DI*), and transiently reduced speed (*Figure 1DII, III*). Furthermore, photo-depolarization of AS MNs evoked body contraction, though contraction was reduced when compared to ChR2 activation of all VNC cholinergic MNs (*Figure 1DIV,V*). These behavioral phenotypes were blue-light dependent, and absent in transgenic animals raised without all-*trans* retinal (ATR), the obligate ChR2 co-factor. As the forward locomotion bending wave propagates from head to tail, we probed how AS MN activity contributes to this propagation. We thus stimulated AS MNs in small segments of the body (anterior, midbody, posterior; *Figure 1EI*). These manipulations neither caused marked disruption of the wave (*Figure 1EII*), nor did they reduce speed. However, they led to a reduction of body length (*Figure 1—figure supplement 2A–C*), most pronounced after stimulation of the anterior segment. As we did not observe strong effects, particularly not on propagation of the undulation wave, we wondered if ChR2-mediated photostimulation may have been too weak to evoke robust AS MN activation. We thus used Chrimson as a more potent optogenetic tool for depolarization (*Klapoetke et al., 2014*; *Figure 1—figure supplement 3*; *Figure 1—video 2*). While effects on body length and bending angles were more pronounced than with ChR2 stimulation, the speed decrease remained transient, and the propagation of the bending wave was still not prominently disrupted or distorted by AS::Chrimson photostimulation. In sum, AS MN depolarization facilitates, but may not play an instructive role in generating the undulatory wave.

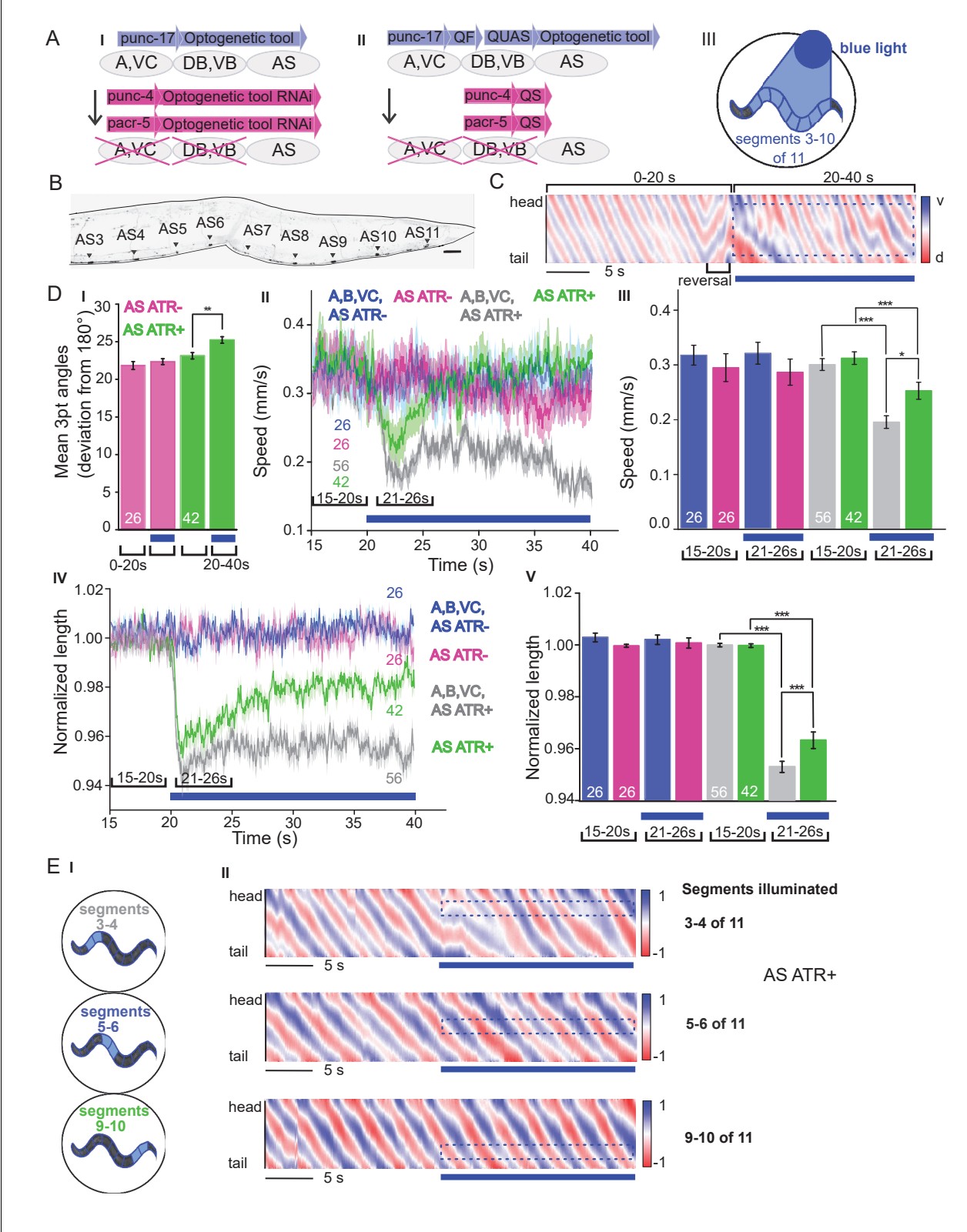

**Figure 1.** Specific photodepolarization of AS MNs via ChR2 leads to body contraction, increased bending angles and reduced speed in freely moving *C. elegans.* (**A**) 'Subtractive' expression and illumination strategy to achieve specific stimulation of AS MNs by optogenetic tools: (I) Silencing of optogenetic tool protein expression in the non-target subsets of MNs by dsRNA; (II) Using the Q system for conditional expression. The transcriptional activator QF binds to the QUAS sequence to induce optogenetic tool expression. The transcriptional inhibitor QS suppresses expression in unwanted

*Figure 1 continued on next page*

*Figure 1 continued*

cells by binding to QF; (III) Selective illumination of the VNC MNs by 470 nm blue light. The body of the worm was divided into 11 segments, of which 3 – 10 were illuminated in animals moving freely on agar plates. (B) Expression pattern of ChR2(H134R)::YFP in AS MNs by the dsRNA subtractive approach; scale bar, 20 μm. See also *Figure 1—figure supplement 1*. (C) Representative body postures kymograph (20 s) of normalized 2-point angles of a 100-point spine, calculated from head to tail of the animal. Positive and negative curvature is represented by blue and red color. Animal expressed ChR2 in AS MNs as in AI and was illuminated after 10 s as in AIII. Blue bar, period of 470 nm illumination. (D) Photodepolarization of AS MNs by ChR2 (in animals raised with ATR): I) Analysis of mean bending angles, before and during the blue light illumination period (as in C). (II, III) Locomotion speed: Mean ±SEM crawling speed of animals before and during blue illumination (blue bar), comparing animals expressing ChR2 in AS MNs or in all types of cholinergic MNs in the VNC, raised in the presence or absence of ATR (III: Group data of mean speed of the animals before (15–20 s) and during (21–26 s) ChR2 photoactivation; IV, (V) Mean ±SEM body length of the animals shown in I, II (V: Group data of the mean length before (15–20 s) and during (21–26 s) photoactivation). (E) Depolarization of subsets of AS MNs in body segments. (I) Scheme of anterior, midbody and posterior segmental illumination; II) Representative body posture kymographs of 2-point angles from head to tail before (20 s) and during ChR2 photoactivation by blue light in the segments of the worm body, corresponding to experiments as in E I). See also *Figure 1—video 1* and *Figure 1—figure supplement 2*. P values *$\leq$0.05; **$\leq$0.01; ***$\leq$0.001. Number of animals is indicated in D. Statistical test in D III and V: ANOVA with Tukey's post hoc test.

DOI: https://doi.org/10.7554/eLife.34997.002

The following video and figure supplements are available for figure 1:

**Figure supplement 1.** Expression of ChR2 can be restricted to AS MNs.
DOI: https://doi.org/10.7554/eLife.34997.003

**Figure supplement 2.** Local AS neuron activation affects body length.
DOI: https://doi.org/10.7554/eLife.34997.004

**Figure supplement 3.** Specific photodepolarization of AS MNs via Chrimson leads to body contraction, increased bending angles and reduced speed in freely moving *C. elegans*.
DOI: https://doi.org/10.7554/eLife.34997.005

**Figure 1—video 1.** Freely moving animal before and during photodepolarization of AS MNs by ChR2 (in animal raised with ATR), blue light = 470 nm, 1.8 mW/mm$^2$.
DOI: https://doi.org/10.7554/eLife.34997.006

**Figure 1—video 2.** Freely moving animal before and during photodepolarization of AS MNs by Chrimson (in animal raised with ATR), red light = 650 nm, 1.8 mW/mm$^2$.
DOI: https://doi.org/10.7554/eLife.34997.007

## AS MN depolarization causes dorsal bias during locomotion by asymmetric BWM activation

We next assessed general effects of AS MN activation on locomotion by analyzing animal tracks. As before, animals expressing ChR2 in all cholinergic neurons slowed (*Figure 1DII, III*) and almost stopped when illuminated (*Figure 2AI*). In contrast, during photostimulation of ChR2 just in AS MNs, animals crawled in circles (*Figure 2AII*; *Figure 1—video 1*), This was due to a bias of head bending towards the dorsal side (*Figure 2AIII–VI*; *Figure 1—figure supplement 3B*), which led to a mild, but significant increase in average bending along the entire body (*Figure 2AV, VI*; this was stronger for AS::Chrimson animals, *Figure 1—figure supplement 3E*), while stimulation of all VNC cholinergic neurons caused deep bending with no dorsal or ventral bias (*Figure 2AIV-VI*). In sum, depolarization of AS MNs contributes to dorso-ventral coordination. Thus, one function of AS MNs may be to facilitate navigation. Unlike the A and B class MNs, the AS MNs have no 'opposing' partner neurons (like VA/DA or VB/DB) and innervate (dorsal) BWMs and inhibitory VD neurons (that innervate ventral muscle). We wondered whether this functional asymmetry affected locomotion during AS MN photostimulation by evoking biased activation of dorsal BWMs. Thus, we used Ca$^{2+}$ imaging (GCaMP3) in BMWs of immobilized animals (*Figure 2BI*). We observed intrinsically evoked activity in muscle cells, that was not synchronized across animals, but which uncovered alternating dorso-ventral activity when the traces of both sides of the animals were aligned to the first dorsal peak (*Figure 2BII, III, IV-VIII*). This activity is likely evoked by the motor nervous system, even in an immobilized animal. When we photostimulated (ChR2) AS MNs (*Figure 2BII, IV-VI,* and *Figure 2—video 1*) in animals raised with ATR, we observed asymmetric responses in the BWM: The Ca$^{2+}$ signal in dorsal muscle cells increased, while it simultaneously decreased in the ventral muscles. Thus, the asymmetry in AS MNs anatomy is reflected also by asymmetry of their functional output.

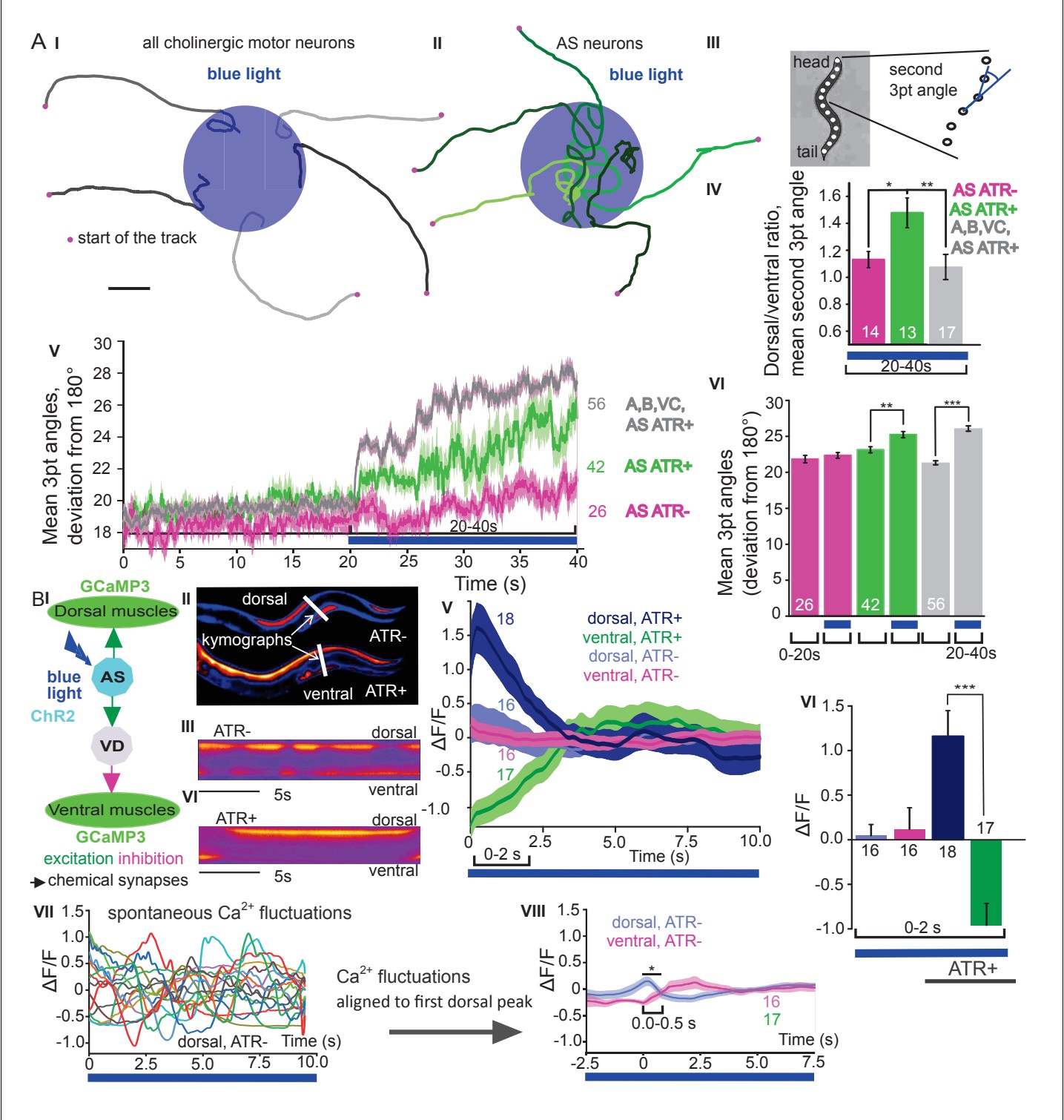

**Figure 2.** Photodepolarization of AS MNs causes transient activation of dorsal and simultaneous inhibition of ventral BWMs, and a dorsal bias in freely crawling animals. A) (I, II) Representative locomotion tracks of freely moving animals (raised with ATR) with ChR2 expressed in all cholinergic MNs (I) or only in AS MNs (II) before (20 s) and during photostimulation (20 s) by 470 nm blue light (indicated by blue shaded area; tracks are aligned such that they cross the blue area at the time of light onset). (III) Schematic showing the thirteen points defining eleven 3-point angles along the spine of the animal. (IV) Mean (±SEM) ratio of dorsal to ventral bending at the 2nd 3-point bending angle in animals expressing AS::ChR2 or ChR2 in all VNC cholinergic neurons during photostimulation (animals raised with and without ATR). (V) Mean (±SEM) time traces of all 3-point bending angles (absolute values) before and during blue illumination (blue bar; ChR2 in AS MNs or in all cholinergic MNs; raised with and without ATR). (VI) Group data of

*Figure 2 continued on next page*

*Figure 2 continued*

experiments in (V), comparing 20 s blue light illumination (blue bar), to the 20 s before illumination. (B) I) AS MNs expressing ChR2 are illuminated by 470 nm blue light, $Ca^{2+}$ signal is recorded in the BWM expressing GCaMP3 (arrows, chemical synapses). (II) Representative snapshots of $Ca^{2+}$ signals in BWM cells during blue light illumination in animals cultivated with and without all-trans-retinal (ATR). Lines indicate regions used to generate kymographs. (III) Representative kymograph of spontaneous $Ca^{2+}$ fluctuations in BWM cells of animal raised without ATR. (IV) Representative kymograph of $Ca^{2+}$ signals in BWM cells in animals cultivated with ATR, in which AS MNs were photostimulated for the entire period. (V, VI) Mean $Ca^{2+}$ signals ($\Delta F/F \pm SEM$) in dorsal and ventral BWM during the first 10 s of illumination (V) in animals raised with and without ATR and group data (VI), quantified during the first 2 s of illumination. (VII) Transients of spontaneous $Ca^{2+}$ signals in dorsal muscles in animal without ATR. VIII) Mean $Ca^{2+}$ signals ($\Delta F/F \pm SEM$) in dorsal and ventral BWM in animals raised without ATR, but aligned to the first dorsal $Ca^{2+}$ spike, showing reciprocity of spontaneous dorso-ventral muscle activity. See also *Figure 2—video 1*. P values *$\leq$0.05; **$\leq$0.01; ***$\leq$0.001; number of animals is indicated. Statistical test: ANOVA with Tukey's post-hoc test.

DOI: https://doi.org/10.7554/eLife.34997.008

The following video is available for figure 2:

**Figure 2—video 1.** $Ca^{2+}$ signal in the BWM expressing GCaMP3 during photodepolarization of AS MNs by ChR2 (in animal raised with ATR), blue light = 470 nm, 1.2 mW/mm$^2$.

DOI: https://doi.org/10.7554/eLife.34997.009

## AS MN ablation disrupts the locomotion pattern

The observed effects indicated an ability of AS MNs to evoke the bending wave during forward locomotion. We can exclude that this was due to ineffective optogenetic stimulation, as even during AS MN stimulation *via* Chrimson, animals performed undulatory forward locomotion (*Figure 1*, *Figure 1—figure supplement 3*; *Figure 1—video 2*). However, given the effects on bending, the AS MNs nonetheless may play an important role in generating locomotion patterns and in navigation. We probed the necessity of AS MNs for locomotion by ablation, as described earlier for other MNs and PINs (*Chalfie et al., 1985*; *Gao et al., 2018*; *McIntire et al., 1993*; *Piggott et al., 2011*). To this end, we used the genetically encoded, membrane targeted (*via* a pleckstrin homology -PH-domain) blue light activated miniature Singlet Oxygen Generator (PH-miniSOG; *Xu and Chisholm, 2016*) and targeted illumination (*Figure 3AI*). Brief illumination of the AS MNs with 470 nm light (2 mW/mm$^2$, 2.5 min) led to visible and quantifiable locomotion defects: Animals with ablated AS MNs retained the ability to move, but crawled with lower speed, increased bending angles and an overall distorted undulation wave along the body, with a highly irregular pattern (*Figure 3AII-IV, Figure 3—figure supplement 1A; Figure 3—video 1*). *Thus, ablation of AS MNs disrupted coordination of the bending wave.*

To further explore how AS MN ablation affects bending, we performed $Ca^{2+}$ imaging in muscle cells of immobilized AS::PH-miniSOG expressing animals, using the red fluorescent $Ca^{2+}$ indicator RCaMP (*Akerboom et al., 2013*), analyzing spontaneous $Ca^{2+}$ signals in ventral and dorsal muscles. Ventral muscles showed $Ca^{2+}$ fluctuations that led to intermediate fluorescence when averaged over many animals, both in the presence, or after the ablation of AS MNs. In contrast, the dorsal muscle on average showed higher fluorescence that, upon AS MN ablation, was significantly reduced compared to ventral muscle (*Figure 3AV-VII*). This emphasizes that AS MNs provide biased dorsal muscle drive.

## Chronic hyperpolarization of AS MNs eliminates $Ca^{2+}$ activity in dorsal BWMs

Photodepolarization of AS MNs caused reciprocal effects on $Ca^{2+}$ signals in dorsal and ventral muscles (*Figure 2*). We wondered whether hyperpolarization of AS MNs may have opposite effects. AS MNs form excitatory chemical synapses to dorsal muscle and to VD MNs (the latter inhibit ventral muscle), but also gap-junctions (to VA MNs, possibly exciting ventral muscle). Thus, several outcomes are conceivable: (1) Decrease of $Ca^{2+}$ levels in dorsal, and increase in ventral muscles; (2) AS MN hyperpolarization may reduce ventral muscle activity via gap junctions to VA MNs; (3) A mixture of both, possibly even causing oscillations. We thus used the *Drosophila* histamine-gated Cl$^-$-channel HisCl1 (*Pokala et al., 2014*), as a hyperpolarizing tool (*Figure 3BI*). Since *C. elegans* has no endogenous histamine receptors, HisCl1 can be specifically activated using histamine. First, we incubated animals expressing HisCl1 in AS MNs with histamine, and compared them to controls not incubated with histamine (*Figure 3BII*; *Figure 3—figure supplement 1B*; *Figure 3—video 2*). Animals on

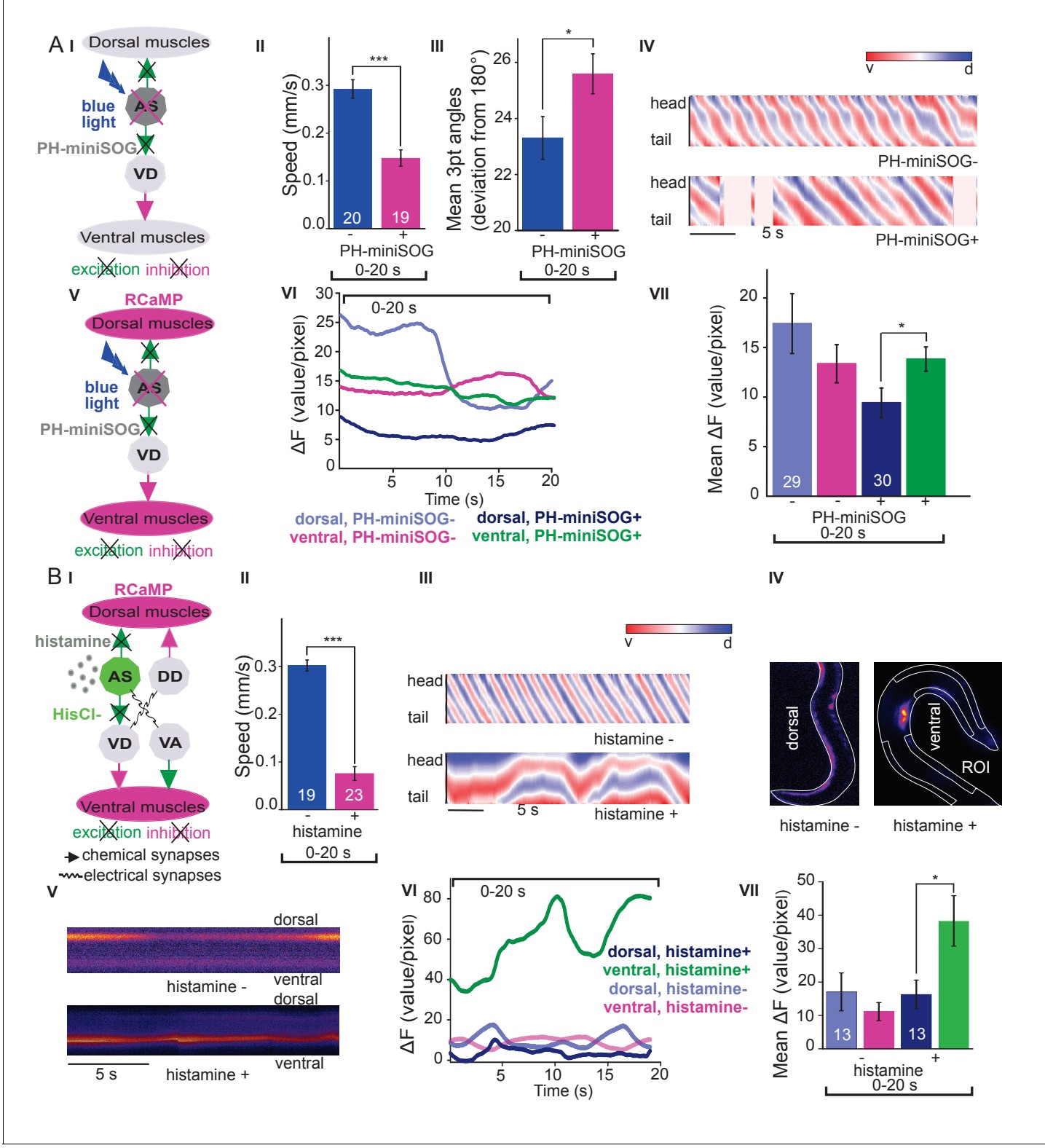

**Figure 3.** Optogenetic ablation and chronic hyperpolarization of AS MNs disrupts the locomotion pattern. (**A**) I) Schematic of optogenetic ablation of AS MNs by PH-miniSOG and connectivity to relevant cell types (arrows, chemical synapses; curved lines, electrical synapses). Quantification of mean ±SEM speed (II) and bending angles (III) of animals without or with expression of PH-miniSOG (*via* the Q system) in AS MNs, following 150 s of blue light exposure and 2 hr resting period. IV) Representative body posture kymographs (as in *Figure 1C*) of wild type animal (upper panel) and animal expressing PH-miniSOG after photoactivation (lower panel). V) Schematic of Ca$^{2+}$ imaging in BWM (RCaMP fluorescence) during PH-miniSOG ablation.
*Figure 3 continued on next page*

Figure 3 continued

VI) Representative transients of Ca²⁺ signaling in dorsal and ventral muscles after AS MN ablation by PH-miniSOG. VII) Mean Ca²⁺ signals (ΔF/F ± SEM) in dorsal and ventral BWM of animals without or with expression of PH-miniSOG. See also *Figure 3—figure supplement 1A* and *Figure 3—video 1*. (B) I) Schematic of Ca²⁺ imaging in BWM (RCaMP fluorescence) during hyperpolarization of AS MNs by HisCl1 (expressed in AS MNs *via* the Q system), and connectivity to relevant cell types (see also AI; note that this simplified diagram reflects cell type connectivity but does not accurately reflect connections between individual cells). (II) Mean ±SEM speed of freely moving animals on agar dishes without and with 10 mM histamine. (III) Representative body posture kymographs of animals freely moving on agar without (upper) or with 10 mM histamine (after 240 s incubation; lower panel). (IV) Representative fluorescent micrographs of Ca²⁺ activity in the BWM of animals mounted on agar slides without (left) or with 10 mM histamine (after 240 s incubation; right panel). (V) Representative kymographs (20 s) of Ca²⁺ activity in dorsal and ventral BWM of animals as in IV. (VI) Representative Ca²⁺ activity in dorsal and ventral BWM from animals as (shown in IV, V. VII) Mean ±SEM fluorescence of dorsal and ventral BWM as in VI. See also *Figure 3—figure supplement 1B* and *Figure 3—video 2*. P values *≤0.05, ***≤0.001; number of animals indicated in AII, VII; BII, VII. Statistics: T-test for AII, III and B II; ANOVA with Tukey's post-hoc test for A VII and BVII.
DOI: https://doi.org/10.7554/eLife.34997.010

The following video and figure supplement are available for figure 3:

**Figure supplement 1.** Optogenetic inactivation and HisCl1-induced hyperpolarization of AS MNs affects locomotion speed and bending angles.
DOI: https://doi.org/10.7554/eLife.34997.011
**Figure 3—video 1.** Freely moving animal, 2 hr after optogenetic ablation (150 s) of AS MNs by PH-miniSOG.
DOI: https://doi.org/10.7554/eLife.34997.012
**Figure 3—video 2.** Freely moving animal expressing HisCl1 after 4 min exposure on 10 mM histamine.
DOI: https://doi.org/10.7554/eLife.34997.013

histamine plates moved significantly slower (ca. 75% reduction) than animals without histamine, demonstrating that AS MNs are actively involved in promoting locomotion (however, this manipulation also affects other cholinergic neurons outside the VNC; see below). To analyze the possible reason for the reduced speed, we analyzed the crawling body postures (*Figure 3BIII*). Histamine exposure strongly disturbed the propagation of the body wave, leading to very slow and irregular movement and frequent directional changes. We assessed the effects of constant AS MN hyperpolarization on muscle physiology and activity again via RCaMP, in immobilized animals (*Figure 3BIV*), either without or with histamine. Consistent with the dorsal innervation of muscles by AS MNs, intrinsically evoked Ca²⁺ activity in animals with hyperpolarized AS MNs was observed only in ventral BWM, and animals showed ventral bending. Over time, on histamine, ventral Ca²⁺ fluctuations had much higher amplitude than those in dorsal muscle, while animals without histamine showed comparable fluctuations in both dorsal and ventral muscles, which were of the same low amplitude as in dorsal muscle with histamine (*Figure 3BV-VII*). In sum, hyperpolarization of AS MNs inhibits their excitatory signaling to dorsal muscles, and blocks their activation of GABAergic VD motor neurons, which leads to ventral muscle disinhibition. Similar effects are found upon ablation of AS MNs, and both types of manipulation cause a strong bias to uniform ventral muscle activation, which is likely to disrupt propagation of the body wave.

## Acute hyperpolarization of AS MNs induces ventral muscle contraction through disinhibition

Chronic hyperpolarization of AS MNs by HisCl1 lacks temporal resolution, and, due to the expression from the *unc-17* promoter, despite our 'subtractive' expression, hyperpolarization of head and tail neurons could affect the outcome of these experiments. To avoid inhibition of these neurons, we looked for a potent hyperpolarizing optogenetic tool, enabling to use selective illumination for specific AS MN inhibition. We thus used the natural Cl⁻-conducting anion channelrhodopsin (ACR1), which causes strong (shunting) inhibition upon illumination (*Sineshchekov et al., 2015*; *Bergs et al., 2018*) *Figure 4A*).

Acute, ACR1-induced photo-hyperpolarization of all cholinergic neurons in freely moving animals strongly reduced crawling speed (to below 20%) and essentially stopped locomotion (*Figure 4B,C*). When we restricted expression of ACR1 and illumination to the AS MNs, we observed a similar reduction of speed, though not as pronounced (to ca. 35% of the initial speed; *Figure 4B*). Controls (animals raised without ATR) showed no change in locomotion speed. These results, together with the HisCl1 experiments may suggest that the speed reduction was caused by a lack of ACh release from AS MNs to dorsal muscles. As this should cause partial relaxation of the body, we analyzed

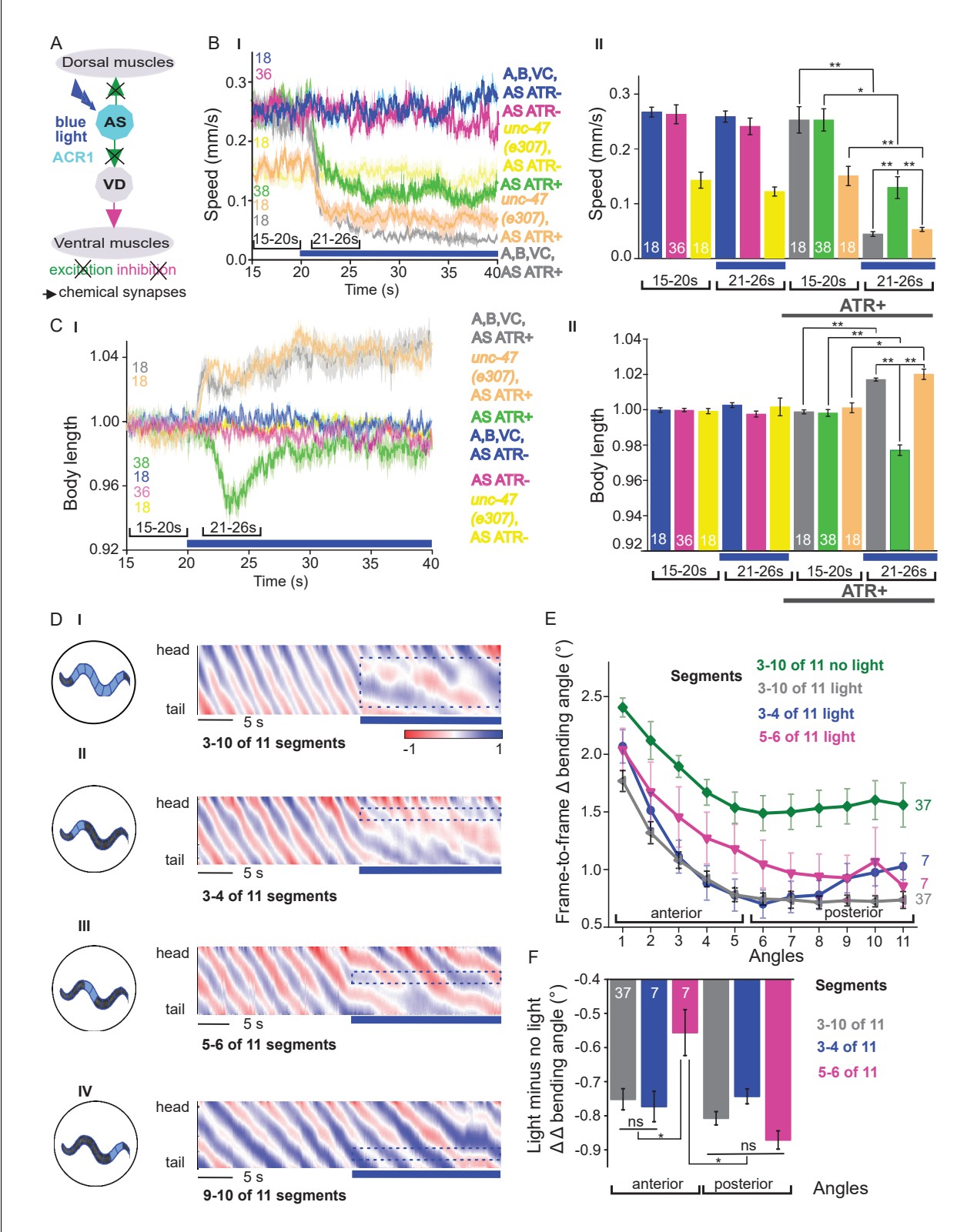

**Figure 4.** Acute optogenetic hyperpolarization of AS MNs ceases locomotion, causes disinhibition of ventral BWM via GABAergic VD MNs, and blocks propagation of the locomotion body wave. (**A**) Schematic of experiment; hyperpolarization of AS MNs by the ACR1 anion channel rhodopsin activated by 470 nm blue light (arrows, chemical synapses). (**B**) Time traces (I) and group data quantification (II) of mean ±SEM speed before (15 – 20 s) and during (21 – 26 s) blue illumination (indicated by blue bar). Compared are strains expressing ACR1 in all VNC cholinergic neurons, or in AS MNs only

*Figure 4 continued on next page*

*Figure 4 continued*

(*via* the Q system), in wild type or *unc-47(e307)* mutant background, raised in the presence or absence of ATR, as indicated. (**C**) Time traces (I) and group data quantification (II) of mean ±SEM body length of the animals shown in B. (**D**) Hyperpolarization of AS MNs in all (I), in the anterior (II), middle (III) and posterior (IV) segments of the worm body. Representative body postures kymographs of normalized 2-point angles from head to tail in animal expressing ACR1 in AS MNs before and during illumination by blue light in the indicated body segments. (**E**) Mean, absolute difference of bending angles, from one video frame to the next (25 Hz), at each of eleven 3-point angles, for experiments as in (**D**). (**F**) Mean difference of the differential bending angles between dark and illuminated conditions, for the analyses shown in (**E**). Data were averaged for the anterior five or the posterior 6 3-pt bending angles. See also *Figure 4—figure supplement 1* and *Figure 4—videos 2-4*. P values *$\leq$0.05; **$\leq$0.01; number of animals is indicated. Statistics: ANOVA with Tukey's post hoc test.

DOI: https://doi.org/10.7554/eLife.34997.014

The following video and figure supplement are available for figure 4:

**Figure supplement 1.** Local stimulation of AS MNs in different body segments.

DOI: https://doi.org/10.7554/eLife.34997.015

**Figure 4—video 1.** Freely moving animal expressing ACR1 in AS MNs before (20 s) and during ACR1 photoactivation by 470 nm 1.8 mW/mm$^2$ blue light.

DOI: https://doi.org/10.7554/eLife.34997.016

**Figure 4—video 2.** Selective illumination of anterior segment.

DOI: https://doi.org/10.7554/eLife.34997.017

**Figure 4—video 3.** Selective illumination of midbody segment.

DOI: https://doi.org/10.7554/eLife.34997.018

**Figure 4—video 4.** Selective illumination of posterior segment.

DOI: https://doi.org/10.7554/eLife.34997.019

body length: For animals expressing ACR1 in all cholinergic neurons, we observed a prominent body elongation, in line with the absence of all excitatory (cholinergic) transmission to muscle (*Figure 4C*). However, hyperpolarization of only the AS MNs led to partial and transient body contraction (*Figure 4C*; *Figure 4—video 1*). This might be explained by synaptic connections of AS MNs to the GABAergic VD MNs: Hyperpolarization of AS MNs would reduce excitation of VD MNs, which in turn would cause dis-inhibition of muscle cells, and thus contraction. To test this, we repeated the experiment in *unc-47(e307)* mutants, lacking the vesicular GABA transporter, and thus GABAergic transmission. Consistently, *unc-47* mutants showed relaxation instead of contraction of BWMs (*Figure 4C*). Body wave propagation was strongly attenuated, as for the analogous experiment using HisCl1 in AS MNs; however, using ACR1, this was induced within 2 – 3 s of illumination. Last, we probed if local AS MN inhibition (i.e. in anterior, midbody or posterior neurons) would block the propagation of the wave posterior from this point (*Figure 4D*). This was the case: In about half of the animals tested, inhibition of anterior and midbody AS neurons hindered propagation of the wave to the posterior part of the body, leading to dragging behind of the tail region (*Figure 4D*; *Figure 4—video 2-4*). Analyses of the extent of movement in individual body segments showed that a reduction of movement was also found in the head region, however, this was more pronounced toward the posterior, particularly when the midbody AS neurons were inhibited (*Figure 4E*; *Figure 4—figure supplement 1A* shows how the eleven 3-point angles analyzed correspond to illuminated body segments): The extent of reduction in body movement in the anterior part of the animal was significantly smaller than the change in posterior body movement (*Figure 4F*). Animals also showed a reduction of speed, though not as pronounced as when all AS MNs were hyperpolarized, and length was not affected (*Figure 4—figure supplement 1A*). When the posterior segment was hyperpolarized, no obvious effects were observed. In sum, AS MNs are required for antero-posterior propagation of the body wave.

## AS MN activity oscillates during crawling, and correlates with body bends

Measuring Ca$^{2+}$ transients in the ventral cord MNs during locomotion revealed higher activity states for B- and A-type MNs during forward and backward locomotion, respectively (*Haspel et al., 2010*; *Kawano et al., 2011*; *Qi et al., 2013*). Correlation of Ca$^{2+}$ traces in AS MNs with dorsal body bends was previously shown in freely crawling animals (*Faumont et al., 2011*). Considering the unique situation of AS MNs, i.e. coupling with both forward and backward command interneurons, we

wondered if AS MNs would maintain equal activity during both locomotion states. Thus, we measured $Ca^{2+}$ transients in AS6 and AS7 in moving animals.

AS6 and AS7 showed oscillatory activity during locomotion, which was correlated with the change of body bends. During forward crawling, (anterior) AS6 activity preceded (posterior) AS7 activity by about 1–2 s (*Figure 5A,B*; *Figure 5—video 1*). To understand if AS MN $Ca^{2+}$ transients are related to the locomotion body wave, we measured the angle defined by the position of AS6, the vulva, and AS7. We then performed cross-correlation analysis (for individual undulations, i.e. full periods of the bending wave) of the $Ca^{2+}$ signal in AS6 or AS7 and the respective bending angle at the given time (*Figure 5D*; *Figure 5—video 1*). Here, the $Ca^{2+}$ signal in AS6 preceded the maximal bending at the vulva by about 2 s, while the signal in AS7 coincided (correlation coefficients were ~0.30–0.35). Thus, the wave of activity in AS MNs appears to travel antero-posteriorly at the time scale of the undulatory wave. Note the animals crawl under a cover slip, slowing down locomotion; in animals moving freely on agar, the wave oscillates with ca. 0.5 Hz (*Gao et al., 2018*), while here, the delay of two maxima of undulation is ca. 3 – 4 s (*Figure 5B*). We also measured cross-correlation between the $Ca^{2+}$ signals in the AS6 and AS7 neurons (coefficient ~0.28). During reversal periods (we only included reversals of at least 10 s; *Figure 5C*), there was weaker correlation (coefficient ~0.15–0.3) of AS6 and AS7 $Ca^{2+}$ signals with the vulva bending angles, or with each other, and there was a smaller time lag between these signals (~1 s, *Figure 5E*). The cross-correlation of $Ca^{2+}$ signals of the neurons and the bending angle were not different between forward and reversal locomotion (*Figure 5F*). The peaks of AS6 and AS7 $Ca^{2+}$ signals did not reveal any difference between forward and reverse movements (*Figure 5G*). In sum, AS MNs showed oscillatory activity that was correlated with body bends during both forward and backward crawling.

## AS MNs integrate signaling from both forward and backward PINs

The PINs AVA, AVD, and AVE connect to the DA and VA MNs, and induce reversals and backward locomotion. Conversely, the PINs AVB and PVC are connected to the DB and VB forward MNs and mediate forward locomotion (*Chalfie et al., 1985*; *Chronis et al., 2007*; *Kawano et al., 2011*; *Piggott et al., 2011*; *Qi et al., 2013*; *Wicks et al., 1996*). Endogenous (*Kawano et al., 2011*) as well as stimulated activity of the PINs modulates activity of A- and B-type MNs (*Gao et al., 2015*; *Gao et al., 2018*; *Liu et al., 2017*).

The AS MNs are postsynaptic to both backward (synapse number: AVA - 63, AVE - 7) and forward PINs (AVB - 13, PVC - 2). This may suggest a bias of AS MNs for backward locomotion; however, synapse number is not the only determinant of synaptic weight. No chemical synapses are known from AS MNs towards the PINs, yet, there are 37 gap junctions reported between AVA and the AS MNs as well as five gap junctions between AVB and the AS MNs. Electrical synapses could mediate anterograde as well as retrograde signaling between AS MNs and PINs (*Chen et al., 2006*; *Varshney et al., 2011*; *White et al., 1986*). To assess whether depolarization of AVA and AVB would lead to observable and/or different $Ca^{2+}$ responses in the AS MNs, we generated strains expressing ChR2 in the PINs and GCaMP6 in AS MNs (*Figure 6AI*): One strain specifically expressed ChR2 in AVA (*Schmitt et al., 2012*) and another strain expressed ChR2 from the *sra-11* promoter in AIA, AIY, and AVB neurons, of which only AVB has direct synaptic connections to AS MNs. The respective animals were photostimulated and $Ca^{2+}$ transients were measured in AS3 (anterior) and AS8 (posterior) MNs of immobilized animals, raised either in absence or presence of ATR (i.e. without and with functional ChR2). Stimulation of AVA or AVB both resulted in a steady, synchronous increase of the $Ca^{2+}$ signal in the AS3 neuron; however, no increase was observed in animals raised without ATR (*Figure 6AII-IV; Figure 6—video 1, 2*). A similar increase was found in the AS8 neuron, and both AS3 and AS8 showed a synchronized increase of activity (*Figure 6—figure supplement 1A,B*). Thus, signaling from both forward and backward PINs is excitatory to the AS MNs. While AVA depolarization evoked a response of comparably low, and AVB depolarization caused a response of significantly higher amplitude (20 vs. 40% ΔF/F after 3 s, respectively; *Figure 6AIV*), this difference is probably not meaningful, as expression levels of ChR2 in AVA vs. in AVB likely differ.

Inequality in regulated behavior based on imbalances in wiring has been observed for the PINs and A- and B-class MNs (*Kawano et al., 2011*; *Roberts et al., 2016*), where AVA coupling to A-type MNs via gap junctions is rectifying towards AVA (*Liu et al., 2017*). We thus assessed the role of the electrical synapses between the PINs and the AS MNs. AS MNs express UNC-7 and INX-3 as the sole innexins. INX-3 is widely expressed in multiple tissues, and AVA and AVB also express UNC-

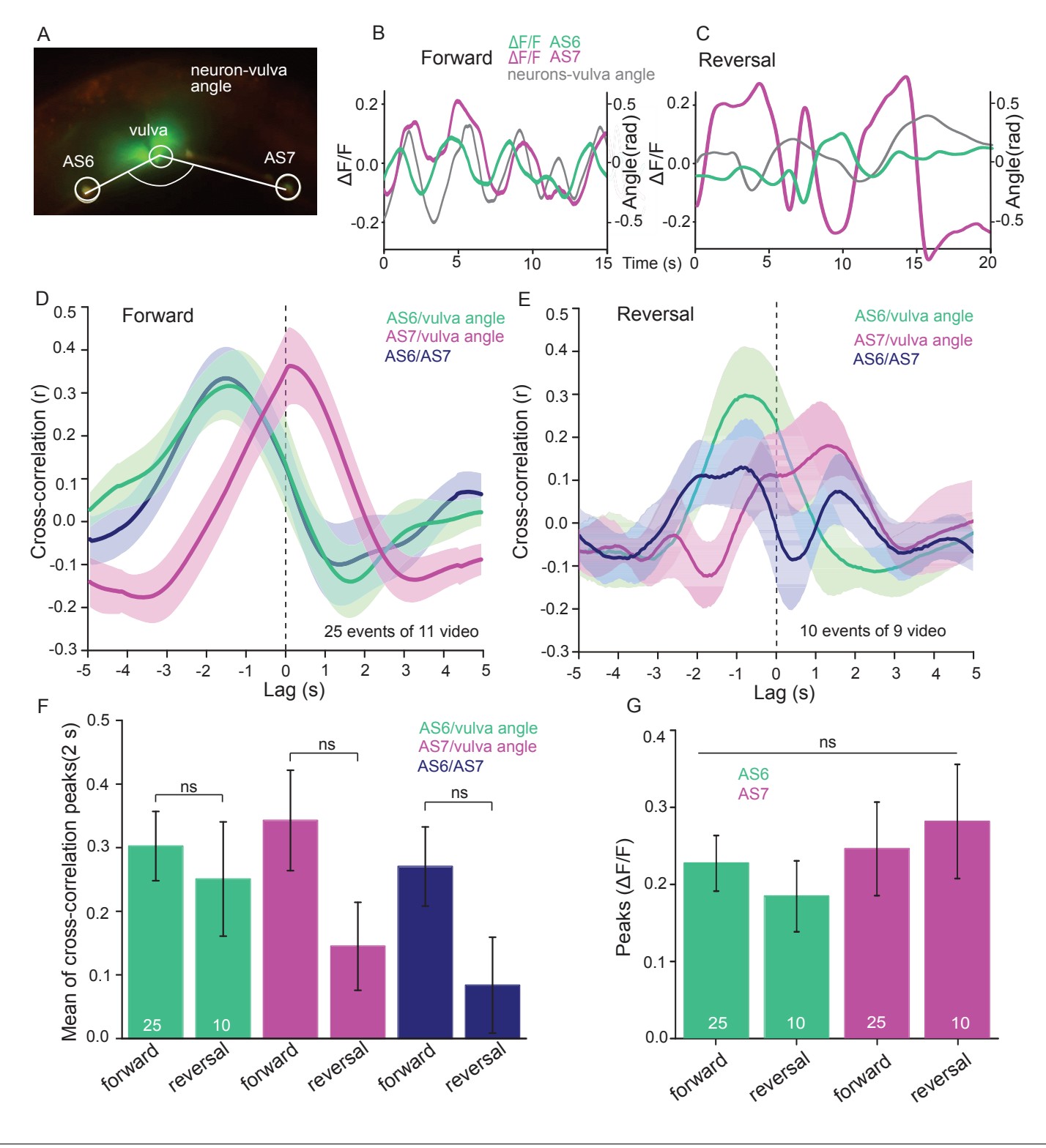

**Figure 5.** AS MNs show oscillatory Ca$^{2+}$activity in moving animals. (**A**) Fluorescent micrograph (merged channels) of the vulva region, showing red (mCherry) and green (GCaMP6), expressed in AS MNs (with the use of the Q system), and GFP, expressed in vulva muscles. Angle between vulva and the two flanking AS6 and AS7 neurons is indicated. (**B, C**) Representative analysis of time traces of Ca$^{2+}$ signals (ΔF/F) in AS6 and AS7, as well as the angle defined by the vulva and the two neurons during crawling (B, 15 s, forward; C, 20 s, reverse). (**D, E**) Cross-correlation analysis (mean ± SEM) of single periods of the body wave (5 s each) for each of the AS6 and AS7 GCaMP6 signals with the vulva angle, as well as for the two Ca$^{2+}$ signals, during

*Figure 5 continued on next page*

*Figure 5 continued*

forward (D) or backward (E) locomotion. (F) Comparison of the mean cross-correlation peaks (during the 2 s centered on the peak) of the fluorescence of AS6 or AS7 and the neuron/vulva angle, or between AS6 and AS7 neurons, for forward or reverse locomotion. (G) Comparison of the peak $Ca^{2+}$ signals (mean ± SEM) in AS6 and AS7, during forward or reverse locomotion, respectively. See also *Figure 5—video 1*. Number of animals is indicated in D-G. Statistical test: ANOVA with Tukey's post-hoc test.

DOI: https://doi.org/10.7554/eLife.34997.020

The following video is available for figure 5:

**Figure 5—video 1.** Moving animal expressing GCaMP6 and mCherry in the AS MNs while the animal is being automatically tracked *via* the GFP marker in vulva muscles.

DOI: https://doi.org/10.7554/eLife.34997.021

7 (*Altun et al., 2009*; *Starich et al., 2009*). Thus, we used an *unc-7(e5)* null mutant, in which no electrical coupling should occur between AS MNs and AVA or AVB, and repeated the above experiments (*Figure 6BI-IV*). This showed similar depolarization of AS MNs following AVB stimulation, while activity of the AS MNs did not increase significantly with ATR. Thus, gap junctions may contribute particularly to AVA-AS MN signaling.

## Retrograde electrical signaling from AS MNs depolarizes AVA but not AVB interneurons

To explore whether there is also retrograde signaling from AS MNs to PINs via gap junctions, we perfomed the reciprocal experiment, i.e. we tested if AS MN photostimulation could lead to depolarization of the PINs. We crossed a transgene expressing ChR2 in the AS MNs with strains expressing the ratiometric $Ca^{2+}$ indicator cameleon (bearing CFP and YFP moieties; *Miyawaki et al., 1997*) in the PINs (driven by *sra-11* and *nmr-1* promoters, for expression in AVB and AVA, respectively, a kind gift by M. Zhen; *Figure 6CI*). Both promoters express in several head neurons, yet we could identify AVB and AVA by their position with respect to anatomical landmarks and with respect to other (known) fluorescent neurons. Photodepolarization of the AS MNs (in animals raised in the presence of ATR) caused $Ca^{2+}$ transients in AVA ($\Delta R/R \sim 18\%$), but had no significant effect on $Ca^{2+}$-activity in AVB interneurons (*Figure 6CII-IV*). The latter is in line with the much smaller number of gap junctions between AS MNs and AVB. A small ($\Delta R/R \sim 9–10\%$), insignificant increase of the ratio of the CFP/YFP signal was observed in the control animals raised without ATR, in both AVA and AVB (likely due to uneven bleaching of CFP vs. YFP). In the *unc-7* gap junction mutant background, the $Ca^{2+}$ signal ($\Delta R/R \sim 11\%$) was now comparable to the signal observed in the control without ATR, indicating that UNC-7 electrical synapses are responsible for transmission between AS MNs and AVA. For AVB, we did not observe any significant effect in the *unc-7(e5)* mutant. Similar observations were made in L4 larvae, which we tested to rule out any differences caused by altered expression levels due to developmental changes (data not shown).

## Discussion

Movement by undulations is remarkably effective across scales and in a variety of environments (*Cohen and Sanders, 2014*). Despite the diversity of their anatomy, the nervous systems of distantly related organisms may adopt similar strategies to control locomotion by undulations. Based on physiological data we revealed several features of the AS MNs highlighting their function in one of the most studied locomotion circuits, the VNC of *C. elegans*. The main findings of this work are: (1) Depolarization of AS MNs does not disrupt locomotion, but causes a dorsal bias. (2) AS MN hyperpolarization inhibits locomotion and prevents generation and propagation of the undulatory wave. (3) AS MN activity oscillates during both forward and reverse locomotion. (4) AS MNs are stimulated by forward as well as reverse premotor interneurons. (5) AS MNs have functional electrical connections to the backward PIN AVA. Our findings for AS MNs in the *C. elegans* locomotor circuit have parallels in several animal models (see below).

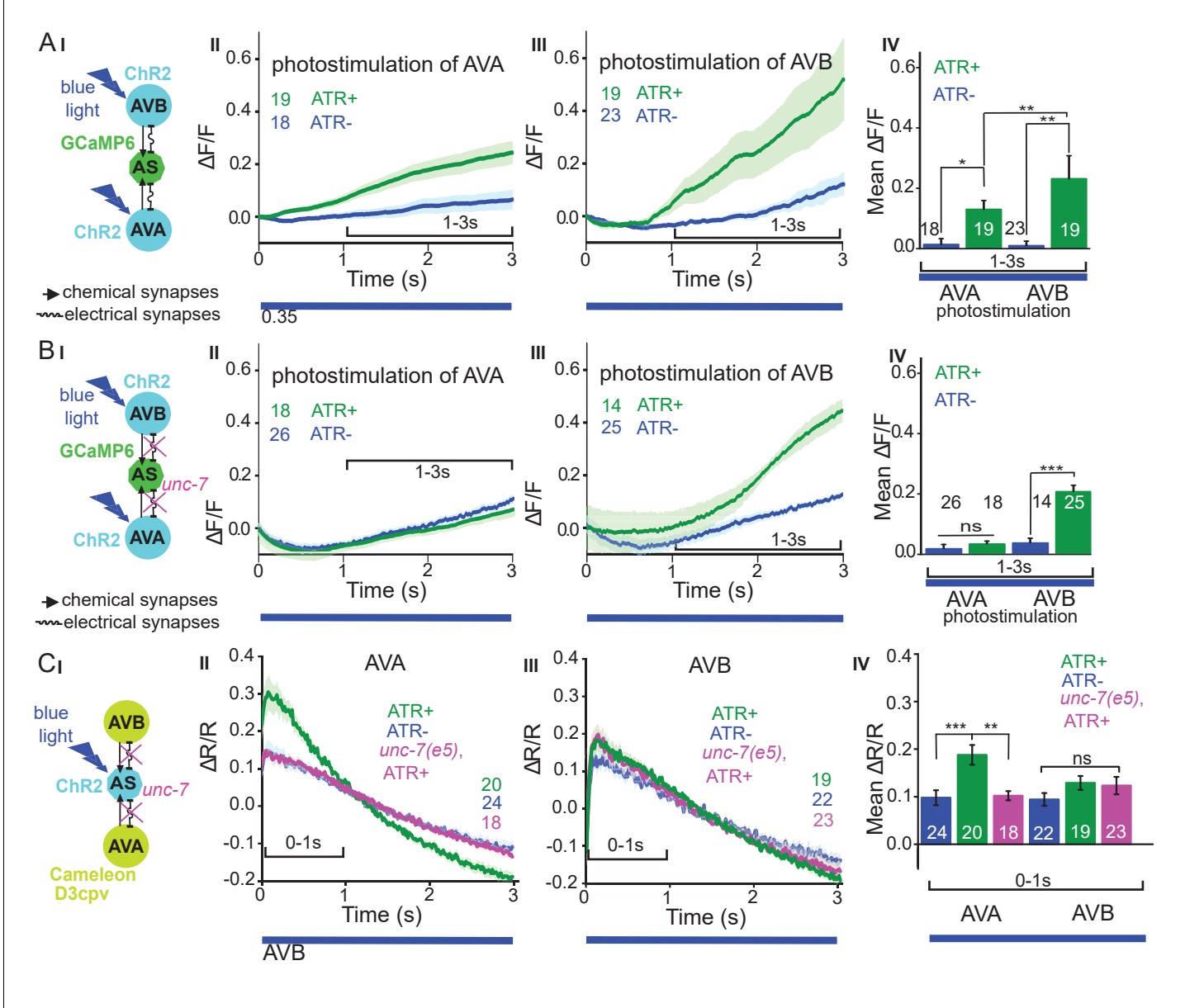

**Figure 6.** Reciprocal and asymmetric mutual activation of AS MNs and forward and reverse PINs, AVB and AVA. (A) (I) Schematic of the experiment for measurement of AS MN Ca²⁺ signals (GCaMP6) during AVB or AVA photodepolarization *via* ChR2 with 470 nm blue illumination (arrows, chemical synapses; curved lines, electrical synapses). (II, III) Time traces of mean (±SEM) Ca²⁺ transients (ΔF/F) in AS MNs during depolarization of AVA (II) and AVB (III) by ChR2, in animals raised in absence or presence of ATR. Brackets indicate time periods used for statistical analysis in IV. (IV) Group data quantification of experiments shown in II and III (for the 1–3 s time period). See also *Figure 6—figure supplement 1* and *Figure 6—video 1, 2*. (B) (I-IV), as in A (I-IV), but in the *unc-7(e5)* gap junction mutant background. (C) (I) Schematic of the experiment for measurement of Ca²⁺ signals (cameleon) in AVB or AVA PINs during AS MN photodepolarization *via* ChR2. (II, III) Mean (±SEM) of Ca²⁺ transients (ΔR/R YFP/CFP ratios) in AVA (II) and AVB (III) during AS MN depolarization, in wild type or *unc-7(e5)* mutant animals, raised in absence or presence of ATR. IV) Group data quantification of experiments in II and III (for the 0–1 s time period). P values *≤0.05; **≤0.01; ***≤0.001; number of animals is indicated. Statistical test: Mann-Whitney U test.

DOI: https://doi.org/10.7554/eLife.34997.022

The following video and figure supplement are available for figure 6:

**Figure supplement 1.** AS MNs are simultaneously activated by photostimulation of the AVA and AVB PINs.

DOI: https://doi.org/10.7554/eLife.34997.023

**Figure 6—video 1.** Ca²⁺ signal in the AS MNs expressing GCaMP6 during photodepolarization of the AVA PIN by ChR2 (in animal raised with ATR), 470 nm blue light, 1.2 mW/mm².

*Figure 6 continued on next page*

*Figure 6 continued*

DOI: https://doi.org/10.7554/eLife.34997.024

**Figure 6—video 2.** Ca$^{2+}$signal in the AS MNs expressing GCaMP6 during photodepolarization of AVB by ChR2 (in animal raised with ATR), 470 nm blue light, 1.2 mW/mm$^2$.

DOI: https://doi.org/10.7554/eLife.34997.025

## AS MNs act in coordination of dorso-ventral bends, antero-posterior wave propagation and possibly forward-backward states

AS MNs occupy a significant part of the VNC circuit (11 of 75 neurons) and two AS MNs are present in each functional segment of the circuit (*Haspel and O'Donovan, 2011*). Yet, in absence of physiological information, they were missing in many models representing the locomotor circuit function in *C. elegans* (*Von Stetina et al., 2005*; *Zhen and Samuel, 2015*). We showed that AS MNs are important for several aspects of locomotion, among them dorso-ventral coordination: Their depolarization activates dorsal BWMs leading to contraction and dorsal bias in freely moving animals, while AS MN hyperpolarization eliminates activity in dorsal BWMs and induces contraction of ventral BWMs through disinhibition. Thus, it is likely that AS MNs counteract neurons providing a ventral bias, e.g. the VA and VB MNs, or even the VC neurons (*Faumont et al., 2011*; *White et al., 1986*). This corresponds to recent computational studies, which predicted a significant role of AS MNs in coordinating dorso-ventral bending (*Olivares et al., 2017*) and in the control of BWMs (*Yan et al., 2017*). Furthermore, AS MNs are active both during forward and reverse locomotion. In line with this, AS MN hyperpolarization disrupts propagation of the antero-posterior body wave. Since the AS MNs connect to both forward and backward PINs, they could play a role in integrating forward and backward locomotion motifs, e.g. by providing an electrical sink (or source; this is more likely for AS-to-AVA signaling, due to the higher number of gap junctions, and in line with our results) for the PINs of the respective opposite direction (*Figure 7A,B*). Similar functions were shown for A-type MNs and AVA (*Kawano et al., 2011*; *Liu et al., 2017*) as well as for V2a interneurons and MNs in zebrafish (*Song et al., 2016*).

AS MNs, as other MN types, innervate only one side of the BWMs (dorsal). However, unlike other MN classes, AS MNs have no obvious class of 'partner' neurons innervating only ventrally. However, there are the VC neurons, which innervate ventral muscle. Yet, there are only six VC neurons, and two of them mainly innervate vulval muscle. AS MNs thus provide asymmetric excitation, which may be required to enable complex regulatory tasks like gradual changing of direction during navigation. Indeed, optogenetic depolarization of AS MNs, resulting in curved locomotion tracks, mimicked the 'weathervane' mode of navigation towards a source of attractive salt (*Iino and Yoshida, 2009*). During locomotion, higher order neurons, that integrate sensory information, might influence the AS MNs to generate this bias to the dorsal side. In the lamprey, lateral bends were caused by asymmetry in stimulation of the mesencephalic locomotor region (*Sirota et al., 2000*), and in the freely moving lamprey even comparatively small left or right asymmetries in activity of the reticulospinal system correspond to lateral turning (*Deliagina et al., 2000*).

Asymmetry in contralateral motifs of complex locomotor circuits is also known from vertebrate spinal cord CPGs in flexor-extensor coordination (*Grillner and Wallén, 2002*). In mice, flexor motor neurons are predominantly active and inhibit extensor motor neurons, which in turn show intervals of tonic activity between inhibitory states, corresponding to flexor bursts (*Machado et al., 2015*; *Rybak et al., 2015*). When comparing numbers of synaptic inputs from *C. elegans* cholinergic MNs to BWM (*Varshney et al., 2011*; *White et al., 1986*), predominance is apparent in excitatory neuromuscular junctions from A- and B-type MNs to ventral muscles (number of synapses: A-type to ventral muscle: 225, to dorsal muscle: 111, B-type to ventral muscle: 228, to dorsal muscle: 58), as well as in the corresponding contralateral synapses to inhibitory DD MNs, which innervate dorsal muscle (VA and VB to DD: 180 and 194 synapses, DA and DB to DD: 8 and 29 synapses). Therefore, tonic activity of B- or A-type MNs would be expected to generate a bias towards ventral bending, and this could be balanced by excitation of AS MNs. In addition, VC neurons may contribute in counteracting AS MN function (see above). However, the compressed nature of the *C. elegans* nervous system, in which single neurons fulfill multiple tasks that in higher animals are executed by layers of different cells, may not always allow for the direct comparison to vertebrate systems.

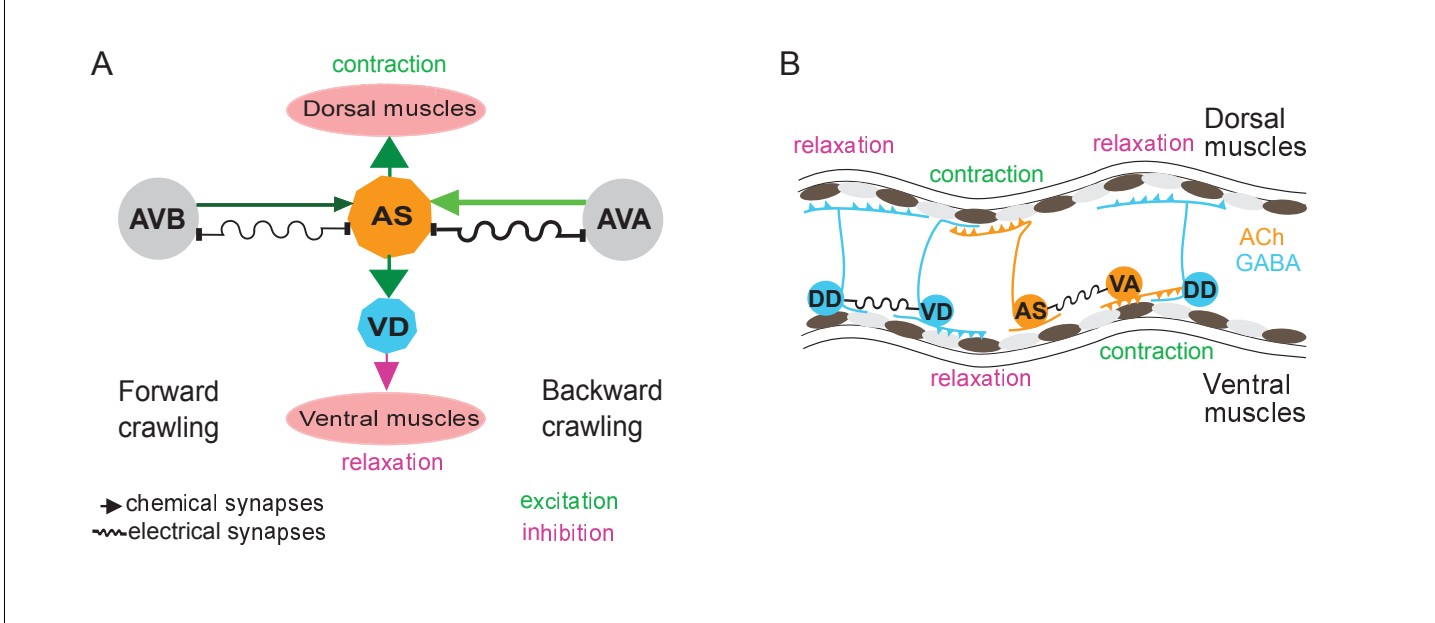

**Figure 7.** Models summarizing findings of this work. (**A**) AS MNs asymmetrically regulate dorso-ventral bending during forward and backward locomotion, by excitatory chemical transmission to dorsal muscles and ventral GABAergic VD MNs, thus causing ventral inhibition. Interconnections (arrows, chemical synapses; curved lines, electrical synapses; thickness of lines indicates relative synapse number as an approximation of synaptic strength; color shade represents strength of functional connections measured in this work) of AS MNs and their other synaptic partners, that is the PINs AVB and AVA, via both chemical synapses from the PINs and (reciprocal) electrical synapses from AS MNs are also shown. Data in this work suggest (chemical) excitatory regulation of AS MNs by AVA and AVB during forward and backward locomotion, respectively, and reciprocal electrical regulation of AVA by AS MNs. (**B**) Interconnections and functional roles of AS MNs and other VNC MNs during the propagation of the undulatory wave along the body. Depolarization (which could be initiated by AVB, not shown here, or by proprioceptive feedback from the adjacent body segment) of AS MNs causes contraction of the dorsal BWMs and simultaneous relaxation of ventral BWMs through the excitation of VD MNs. This phase is followed by contraction of ventral BWMs, e.g. through the electric coupling of AS and VA MNs, and relaxation of the dorsal BWMs through VD-DD electrical coupling or VA-DD chemical synapses. Cholinergic (orange) and GABAergic (blue) cell types are indicated. Antero-posterior localization of cell bodies and connectivity to other cell types are arbitrary.

DOI: https://doi.org/10.7554/eLife.34997.026

The groups of forward (AVB, PVC) and backward PINs (AVA, AVD, AVE), respectively, are synchronized (*Prevedel et al., 2014*), inhibit each other and change their state stochastically (*Pierce-Shimomura et al., 2008*; *Roberts et al., 2016*). Recently, ability of MNs to modulate activity of PINs was shown in several animal models: for B-type MNs changing the inhibitory chemical transmission of AVB to AVA in *C. elegans* (*Kawano et al., 2011*; *Liu et al., 2017*), for MNs regulating the frequency of crawling in *Drosophila* (*Matsunaga et al., 2017*), for MNs affecting activity of the excitatory V2a neurons in zebrafish (*Song et al., 2016*), and, in mice, such activity was suggested for MNs, changing the frequency of rhythmic CPG activity after stimulation by rhodopsins (*Falgairolle et al., 2017*). Positive feedback from MNs required the function of gap junctions, coupling between MNs and PINs in all these systems. Our data suggest that AS MNs have electrical feedback to the backward PIN AVA. No signal was seen in AVB, yet expression of the $Ca^{2+}$ indicator cameleon was less in AVB, which may have influenced the outcome (see also *Kawano et al., 2011*). If the AS neurons would provide inhibition in the context of the free moving animal, e.g. as an electrical sink for AVA, AS MN activity may exert a bias to promote the forward locomotion state.

## AS MNs as part of CPGs in the ventral nerve cord?

CPGs are dedicated neural circuits with intrinsic rhythmic activities (*Grillner, 2006*; *Guertin, 2012*). In many organisms including those showing undulatory movement (e.g. leech, lamprey), series of CPGs are distributed along the length of the body in locomotor neural circuits (*Kristan et al., 2005*; *Mullins et al., 2011*). In *C. elegans*, the bending wave can be generated even in the absence of all PINs (*Gao et al., 2018*; *Kawano et al., 2011*; *Zheng et al., 1999*) as well as in absence of

GABAergic MNs (*Donnelly et al., 2013*; *McIntire et al., 1993*). The existence of series of CPGs in the *C. elegans* VNC was discussed for a long time, and single neurons or small groups of neurons were suggested (*Cohen and Sanders, 2014*; *Gjorgjieva et al., 2014*; *Zhen and Samuel, 2015*). B-type MNs are able to propagate the bending wave posteriorly. The bending may originate from a possible head CPG that generates head oscillations (*Hendricks et al., 2012*; *Shen et al., 2016*), simply by proprioceptive coupling (*Wen et al., 2012*), though recent work showed that also gap junctions are involved (*Xu et al., 2018*). Recently, pacemaker properties of the posterior A-type MN, DA9, were revealed during backward locomotion, that are based on the activity of a P/Q/N-type $Ca^{2+}$ channel (*Gao et al., 2018*), and local oscillators were shown to function in the VNC also for forward locomotion (*Fouad et al., 2018*). Computational modelling of repeating units of the VNC (*Haspel and O'Donovan, 2011*), based on connectome data, identified a dorsally oriented sub-circuit consisting of AS, DA, and DB MNs, which could act as a potential CPG (*Olivares et al., 2017*). Our data on recording dorso-ventral and temporally coordinated oscillatory muscle $Ca^{2+}$ transients in immobilized animals may be a consequence of rhythmic activity in the motor nervous system. Yet, whether AS MNs are part of this activity is currently unknown. The oscillations of AS MN $Ca^{2+}$ levels we observed both during forward and reverse locomotion are not per se an indication of pacemaker activity, as they may be affected by proprioceptive feedback or through other motor neurons. We did not observe them in immobilized animals, which would have been a stronger indication of autonomous CPG activity.

## Potential gating properties of AS MNs

Among PINs, AVB and AVA are most important for enabling forward and backward locomotion, respectively (*Chalfie et al., 1985*; *Kato et al., 2015*; *Roberts et al., 2016*). Bistable states with two distinct membrane potentials, i.e. up and down states, that 'gate' activity of the downstream target neurons, were shown for several interneurons including AVA and AVB (*Gordus et al., 2015*; *Kato et al., 2015*; *Mellem et al., 2008*). For MNs, bistability was inferred for the A-, B- and D- types of MNs from direct recordings (*Liu et al., 2014*) as well as from the graded responses in muscles, corresponding to shorter or longer activity bursts in MNs (*Liu et al., 2017*). Further, all-or-none responses in BWM cells, corresponding to spiking neurons as well as to mammalian skeletal muscles, that result from integrating graded excitatory and inhibitory input from MNs, were demonstrated (*Gao and Zhen, 2011*; *Liu et al., 2009*). AS MNs may similarly integrate inputs from forward and backward PINs, or themselves influence the PINs via UNC-7 and/or INX-3 gap junctions, to gate signal propagation in the VNC during forward locomotion, or to couple to A-type MN oscillators (DA9), via the AVA PIN during backward locomotion (*Gao et al., 2018*). In line with this hypothesis, we found ceasing of locomotion when AS MNs were hyperpolarized. Gating neurons that affect rhythmic properties of CPGs are also known for the leech locomotor circuit (*Friesen and Kristan, 2007*; *Mullins et al., 2011*; *Taylor et al., 2000*).

## Conclusions

The previously uncharacterized class of AS motor neurons is specialized in coordination of dorso-ventral undulation bends during wave propagation, a feature maintained by asymmetry in both synaptic input and output. Moreover AS neurons integrate signals for forward and reverse locomotion from premotor interneurons and potentially gate ventral nerve cord circuits and PINs via gap junctions.

# Materials and methods

## Strains and genetics

*C. elegans* strains were maintained under standard conditions on nematode growth medium (NGM) and fed by *E. coli* strain OP50-1 (*Brenner, 1974*). Transgenic lines were generated using standard procedures (*Fire and Pelham, 1986*) by injecting young adult hermaphrodites with the (plasmid-encoded) transgene of interest and a marker plasmid that expresses a fluorescent protein. In some cases, empty vector was included to increase the overall DNA concentration to 150 –200 ng/µl.

The following strains were used or generated for this study: N2 (wild type isolate, Bristol strain), CB5: *unc-7(e5)X*, CB307: *unc-47(e307)III*, CZ16469: *acr-2(n2420)X*; *juEx4768[psra-11::ChR2::yfp]*

(*Qi et al., 2013*), PD4665: *wt; ccls4655[pes-10::GFP;dpy-20+]*, RM2558: *wt; ls[punc-17::GFP-NLS]*, ZM5091: *wt; hpIs190[pnmr-1(short2)-D3cpv; lin-15+]*, ZM5089: *unc-7(e5)X; hpIs190*, ZM5132: *wt; hpIs179[psra-11-D3cpv]*, ZM5136: *unc-7(e5)X; hpIs179* (all ZM strains are kind gift from Mei Zhen), ZX460: *wt; zxIs6[punc-17::ChR2(H134R)::yfp; lin-15+]V*, ZX499: *wt; zxIs5[punc-17::ChR2(H134R)::yfp; lin-15+]X*, ZX1023: *lite-1(ce314)X; zxIs30[pflp-18::flox::ChR2mCherry::SL2::GFP; pgpa-14::nCre; lin-15+]*, ZX1396: *wt; zxIs51[pmyo-3::RCaMP1h]*, ZX2002: *lite-1(ce314)X; zxIs6*, ZX2004: *lite-1(ce314)X; zxEx1016 [punc-4::ChR2_RNAi~sense & antisense~ pmyo-2::mCherry]; zxEx1017[pacr-5::ChR2_RNAi~sense & antisense~; pmyo-3::mCherry]*, ZX2007: *wt; zxIs5; zxEx1016; zxEx1017*, ZX2008: *wt; zxEx1023[punc-17:: QF; pacr-5::QS::mCherry; punc-4::QS::mCherry; QUAS::ACR1::YFP; pmyo-2::mCherry]*, ZX2011: *wt; zxEx1020[punc-17::QF; pacr-5::QS::mCherry; punc-4::QS::mCherry; QUAS::HisCl1::GFP; pmyo-2:: mCherry]*, ZX2012: *lite-1(ce314)X; ccls4655[pes-10::GFP; dpy-20+]; zxEx1021[punc-17::QF; pacr-5:: QS::mCherry; punc-4::QS::mCherry; QUAS::GCaMP6::SL2::mCherry; pmyo-2::mCherry]*, ZX2110: *wt; mdIs[punc-17::GFP-NLS;] zxEx1024[punc-17::QF; pacr-5::QS::mCherry; punc-4::QS::mCherry; QUAS::PH-miniSOG; pmyo-2::mCherry]*, ZX2113: *unc-47(e307)III; zxEx1029[punc-17::QF; pacr-5:: QS::mCherry; punc-4::QS::mCherry; QUAS::ACR1::YFP; pmyo-2::mCherry]*, ZX2114: *wt; zxIs51; zxEx1020*, ZX2127: *lite-1(ce314)X; zxIs30; zxEx1021*, ZX2128: *lite-1(ce314)X; juEx4768; zxEx1021*, ZX2132: *wt; zxIs5; zxEx1016, zxEx1017; zxEx1028[pmyo-3::GCaMP3]*, ZX2212: *lite-1(ce314)X; hpIs179; zxIs6; zxEx1016, zxEx1017*, ZX2213: *lite-1(ce314)X; hpIs190; zxIs6; zxEx1016, zxEx1017*, ZX2217: *unc-7(e5)X; hpIs190; zxIs6; zxEx1016, zxEx1017*; ZX2220: *unc-7(e5); hpIs179; zxIs6; zxEx1016, zxEx1017*, ZX2221: *unc-7(e5)X; zxIs6; zxEx1016, zxEx1017*,ZX2427: *unc-7(e5); juEx4768; zxEx1021*; ZX2428: *unc-7(e5); zxIs30; zxEx1021*; ZX2429: *wt; zxIs51; zxEx1111[punc-17::QF; pacr-5:: QS::mCherry; punc-4::QS::mCherry; QUAS::PH-miniSOG; pmyo-2::mCherry]*; ZX2434: *wt; zxEx1112 [punc-17::QF; pacr-5::QS::mCherry; punc-4::QS::mCherry; QUAS::Chrimson::GFP; pmyo-2::mCherry]* (Schild, Glauser, 2015; pdg274[QUAS::Chrimson::GFP] is a kind gift from Dominique Glauser), ZX2435: *zxEx432[pglr-1::ChR2(C128S)::YFP; lin-15]; zxEx1021*.

## Molecular biology

We used the following promoters:3.5 kb *punc-17* (*Sieburth et al., 2005*), 2.5 kb *punc-4* (*Miller et al., 1992*) and 4.3 kb *pacr-5* (*Winnier et al., 1999*), genomic DNA sequence upstream of the ATG start codon of each gene, respectively. The *punc-4*::ChR2 sense and antisense construct was generated as follows: The p*unc-4* promoter from plasmid pCS139 was subcloned into pCS57 (*Schultheis et al., 2011*) by *Sph*I and *Nhe*I, 1 kb antisense sequence was amplified from the ligated construct pOT1 *punc-4*::ChR2::YFP using primers (5'-GGGGTTTAAACAGCTAGCGTCGATCCATGG-3' and 5'-CCCGCGGCCGCCCAGCGTCTCGACCTCAATC-3') and subcloned into the same construct by *Not*I and *Pme*I to get pOT2. To silence ChR2 expression under the *pacr-5* promoter we used a sense and antisense strands approach (*Esposito et al., 2007*) as follows: The *pacr-5* promoter was amplified from genomic DNA using primers (5'-TTATGATGCGAAAGCTGAATCGAGAAAGAG-3', 5'-CCATGCTTACTGCACTTGCTTCCCATACTTC-3', nested 5'-GGGGCATGCATCGAGAAAGA-GAAGCGGCG-3', 5'-CCCGCTAGCAAAGCATTGAAACTGGTGAC-3') and subcloned into pCS57 with *Sph*I and *Nhe*I to yield pOT3 (*pacr-5*::ChR2::YFP). The sense and antisense strands were amplified from this construct using primers (for the coding region of ChR2: 5'-ATGGATTA TGGAGGCGCCC-3', 5'-CCAGCGTCTCGACCTCAATC-3'; for the promoter sense: 5'-GGCGGA-GAGTAGTGTGTAGTG-3' and 5'-GGGCGCCTCCATAATCCATCAAAGCATTGAAACTGGTGAC-GAG-3'; for the promoter antisense: 5'-GGCGGAGAGTAGTGTGTAGTG-3' and 5'-GATTGAGG TCGAGACGCTGGCAAAGCATTGAAACTGGTGAC GAG-3'; for fusion of sense strand: 5'-GCGG TTTCACGCTCTGATGAT-3' and 5'-CTCAGTGCCACCAATGTTCAA-3'; and for fusion of the anti-sense strand: 5'-GCGGTTTCACGCTCTGATGAT-3' and 5'-GCGCGAGCTGCTATTTGTAA-3').

The pOT6 *punc-17*::QF construct was generated as follows: QF::SL2::mCherry sequence was amplified from plasmid XW08 (a kind gift from Kang Shen and Xing Wei) using primers 5'-CAGGAG-GACCCTTGGATG CCGCCTAAACGCAAGAC-3' and 5'-AGTAGAACTCAGTTTTCTGATGA-CAGCGGCCGATG-3', and subcloned into pCS57 (*Schultheis et al., 2011*) using In-Fusion cloning (Takara/Clontech). The SL2::mCherry fragment was cut out by *Sal*I and *Bbv*CI and 5'-overhangs were filled in with Klenow polymerase (NEB). The pOT8 *pacr-5*::QS::mCherry construct was generated by subcloning the *pacr-5* promoter from pOT3 into vector XW09 (*Wei et al., 2012*) (a gift from Kang Shen and Xing Wei) with *Sph*I and *Nhe*I. The pOT7 *punc-4*::QS::mCherry construct was generated by subcloning the *punc-4* promoter from the pOT1 plasmid into XW09 vector with *Sph*I and *Nhe*I. The

pOT10 pQUAS::Δpes-10::HisCl1::GFP construct was generated by subcloning the sequence encoding HisCl1::GFP from plasmid pNP403 (*Pokala et al., 2014*) (kind gift from Navin Pokala and Cori Bargmann) into the XW12 vector (a gift from Kang Shen and Xing Wei) with *Asc*I and *Psp*OMI. The pOT11 pQUAS::Δpes-10::GCamP6::SL2::mCherry construct was generated as follows: pQUAS::Δpes-10 sequence was amplified from plasmid XW12 using primers 5'-ACAGCTATGACCATGATTACGC-CAAG-3' and 5'-CCCCGCGGCCGCCCAATCCCGGGGATCCTCTA-3', and subcloned into plasmid p*lin-11*::GCaMP6::SL2::mCherry with *Sph*I and *Not*I. The pOT13 pQUAS::Δpes-10::ACR1::YFP construct was generated as follows: ACR1::YFP sequence was amplified from plasmid pAB03 (*punc-17*:: ACR1::YFP; *Bergs et al., 2018*) using primers 5'-CCCCGGCGCGCCATCCATGAGCAGCATCACC-3' and 5'-CCCCGAATTCCTTACTTGTACAGCTCGTCCAT-3', and subcloned into vector XW12 with *Asc*I and *Eco*RI. Plasmid pOT17 (pQUAS::Δpes-10::PH-miniSOG(Q103L)) was generated as follows: PH-miniSOG(Q103L) sequence was amplified from plasmid pCZGY2849 (a gift from Andrew Chisholm: Addgene plasmid 74112) using primers 5'-CCCCGGCGCGCCCTTCGGATCCAGATCTA TGCAC-3' and 5'-TGTACAAGAAAGCTGGGTCG-3') and subcloned into vector XW12 using *Asc*I and *Eco*RI restriction sites. The construct details are available on request.

## Animal tracking and behavioral analysis

For worms moving freely on NGM, locomotion parameters were acquired with a previously described worm tracker (*Stirman et al., 2011*) allowing to precisely target illumination of identified segments of the worm body by a modified off-the-shelf liquid crystal display (LCD) projector, integrated with an inverted epifluorescence microscope. Light power was measured with a powermeter (PM100, Thorlabs, Newton, NY, USA) at the specimen focal plane. Animals used in all the optogenetics experiments were raised in the dark at 20 ℃ on NGM plates with *E. coli* OP50-1 and all-*trans*-retinal. The OP50-retinal plates were prepared 1–2 days in advance by seeding a 6 cm NGM-agar plate with 250 µl of OP50 culture and 0.25 µl of 100 mM retinal dissolved in ethanol. Young adults were transferred individually on plain NGM plates under red light (>600 nm) in a dark room and kept for 5 min in the dark before transfer to the tracker.

For experiments with ChR2 depolarizing MNs (*Figures 1* and *2*), blue light of 470 nm and 1.8 mW/mm$^2$ intensity was used with the following light protocol: 20 s' dark' (referring to no blue light illumination) control, 20 s of illumination, followed again by 20 s dark. The animals' body was divided into 11 segments, of which 3 – 10, 3 – 4, 5 – 6 or 9 – 10 were illuminated, depending on the experiment.

For optogenetic ablation experiments (*Figure 3A*; *Figure 3—figure supplement 1*), AS MNs were ablated in animals expressing PH-miniSOG by 2.5 min exposure to 470 nm light of 1.8 mW/mm$^2$ intensity; segments 3 – 10 out of 11 were illuminated. Animals were analyzed after a 2 hr resting period for 60 s without illumination. Wild type worms were used as a control with the same illumination protocol. Ablation was verified by fluorescence microscopy in strain ZX2110 expressing green fluorescent protein (GFP) in all cholinergic neurons, in addition to PH-miniSOG in AS MNs.

For experiments of AS MN hyperpolarization using the histamine-gated Cl$^-$-channel HisCl1 (*Figure 3B*; *Figure 3—figure supplement 1*), worm locomotion was measured on NGM plates with 10 mM histamine 4 min after transfer from plates without histamine, for 60 s without illumination. The same strain on NGM without histamine served as a control.

For experiments in which MNs were hyperpolarized with natural Cl$^-$-conducting anion channel rhodopsin (ACR1; *Figure 4*), due to the high operational light sensitivity of the channel, the system was modified as described (*Steuer Costa et al., 2017*). An additional band pass filter (650 ± 25 nm) was inserted in the background light path and a mechanical shutter (Sutter Instrument Company, Novato, USA), synchronized to the light stimulation, was placed between projector and microscope. Control animals were tested for the background light stimulation and showed no response. The light stimulation protocol was 20 s without illumination, 20 s in 70 µW/mm$^2$ 470 nm light and 20 s without illumination. The worms' body was divided into 11 segments, and segments 3 – 10, 3 – 4, 5 – 6 or 9 – 10 were illuminated, respectively. As the experiment in *unc-47(e307)* background was performed with a different transgene injected into *unc-47(e307)* mutants, we tested the extrachromosomal array after outcrossing into wild type background, where it evoked contraction of the animals, as expected (*Figure 4—figure supplement 1*).

For experiments in which MNs were depolarized with Chrimson (*Figure 1—figure supplement 1*) red light of 650 nm and 1.8 mW/mm$^2$ intensity was used with the following light protocol: 20

s 'dark' (referring to no red light illumination) control, 20 s of illumination, followed again by 20 s dark).

Tracks were automatically filtered to exclude data points from erroneously evaluated movie frames with a custom-made workflow in KNIME (KNIME Desktop version 3.5, KNIME.com AG, Zurich, Switzerland; *Warr, 2012*). Our constraints were that animals do not move faster than 1.25 mm/s and their length does not show a discrepancy above 25% to the mean first five seconds of the video. Videos were excluded from analysis when more than 15% of the data points had to be discarded by our constraints. Behavior data passed the Shapiro-Wilk normality test.

For determination of the ratio of dorso-/ventral angles (*Figure 1BIII, IV*), the second out of 11 three point angles, measured from head to tail, was registered for animals for which the vulva position was previously indicated by manually indicating this to the software. For each track, values of the second three-point angle were averaged for dorsal and ventral bends individually, and the ratio was calculated.

To calculate the frame-to-frame difference of bending angles (*Figure 4E*), data on each of eleven 3-point angles were extracted, smoothed by running an average of 15 frames, and the Δ of absolute values between two subsequent frames were calculated and averaged for before and during illumination conditions for each angle. The light – no light Δ Δ of bending angles (*Figure 4F*) were calculated by subtracting the value of the no light from the light condition. They were then averaged for the bending angles 1–5 (anterior) and 6–11 (posterior).

## Body posture analysis

Binarized videos of freely crawling animals were used to segment the animals' body, and analyzed as described earlier (*Hums et al., 2016*; *Stephens et al., 2008*), using a custom MATLAB script (MathWorks, Natick, Massachusetts). Briefly, grey scale worm images were binarized with a global image threshold using Otsu's method (*Otsu, 1979*). Objects encompassing border pixels were ignored and only the largest object was assumed to be the worm. The binary image was further processed (by thickening, removing spur pixels, flipping pixels by majority and filling holes). Worm skeletonization was achieved by thinning to produce an ordered vector of 100 body points and corresponding tangent angles (theta) from head to tail. Images that could not be analyzed or where the skeleton of the animal was unusually small were considered as missing data points. Head and tail assignment was checked manually. The theta angles were smoothened by a simple moving average with a window of 15 centered data points. The mean of these angles was then compared to the Eigenworms computed from previously published data on N2 videos (*Stephens et al., 2008*). The Eigen projections obtained were taken as a measure of worm posture and plotted.

## Fluorescence microscopy

Animals were immobilized in M9 saline with 50 mM $NaN_3$ and mounted on 10% agar pads with polystyrene beads. Images were recorded under 40x magnification on an inverted fluorescence microscope (Axiovert 200, Zeiss, Germany) or with a Zeiss Cell Observer SD, 488 nm excitation laser at 40% power and a Rolera EM-C2 with EM gain of 100, full resolution and 100 ms exposure time. ImageJ tools were used to obtain maximum projections, image straightening and aligning.

## $Ca^{2+}$ imaging

*Microscope setup:* Fluorescence measurements were carried out on an inverted fluorescence microscope (Axiovert 200, Zeiss, Germany) equipped with motorized stage MS 2000 (Applied Scientific Instrumentation, USA) and the PhotoTrack quadrant photomultiplier tube (PMT; Applied Scientific Instrumentation, USA). Two high-power light emitting diodes (LEDs; 470 and 590 nm wavelength, KSL 70, Rapp Optoelektronik, Germany) or a 100 W HBO mercury lamp were used as light sources. A Photometrics DualView-Λ beam splitter was used to obtain simultaneous dual-wavelength acquisition; these were coupled to a Hamamatsu Orca Flash 4.0 sCMOS camera operated by HCImage Live (Hamamatsu) or MicroManager (http://micro-manager.org). Light illumination protocols (temporal sequences) were programmed on, and generated by, a Lambda SC Smart shutter controller unit (Sutter Instruments, USA), using its TTL output to drive the LED power supply or to open a shutter when using the HBO lamp.

## Measurement of Ca$^{2+}$ in muscles and AS MNs in immobilized worms

For measurements of GCaMP3 (*Figure 2*) and RCaMP (*Figure 3B*) in muscles and GCaMP6 in AS MNs (*Figures 5* and *6*), the following light settings were used: GFP/mCherry Dualband ET Filterset (F56-019, AHF Analysentechnik, Germany), was combined with 532/18 nm and 625/15 nm emission filters and a 565 longpass beamsplitter (F39-833, F39-624 and F48-567, respectively, all AHF). ChR2 stimulation was performed using 1.0 – 1.2 mW/mm$^2$ blue light, unless otherwise stated. To measure RCaMP or mCherry fluorescence, 590 nm, 0.6 mW/mm$^2$ yellow light was used. The 2x binned images were acquired at 50 ms exposure time and 20 fps. Animals were immobilized on 2 or 4% M9 agar pads with polystyrene beads (Polysciences, USA) and imaged by means of 25x or 40x oil objective lenses. 5 s of yellow light illumination and 15 s of blue light illumination protocols were used. For RCaMP imaging 20 s yellow light illuminations were used. Measurements of control animals (i.e. raised without ATR, or without histamine) were conducted the same way as for animals kept in the presence of ATR, or exposed to histamine.

Image analysis was performed in ImageJ (NIH). For Ca$^{2+}$-imaging in muscles, regions of interest (ROIs) were selected for half of the BWM cells in the field of view, or around the neuron of interest for Ca$^{2+}$-imaging in AS MNs. Separate ROIs were selected for background fluorescence with the same size. Mean intensity values for each video frame were obtained and background fluorescence values were subtracted from the fluorescence values derived for GCaMP or RCaMP. Subtracted data was normalized to $\Delta F/F = (F_i-F)/F$, where $F_i$ represents the intensity at the given time point and F represents the average fluorescence of the entire trace.

## Measurement of Ca$^{2+}$ in muscles and AS motor neurons in moving animals

Measurements of GCaMP6 and mCherry were performed using the same filter and microscope settings as for immobilized worms. Moving worms were assayed on 1% agar pads in M9 buffer. Tracking was based on the PhotoTrack system (Applied Scientific Instrumentation, USA) that uses the signals from a 4-quadrant photomultiplier tube (PMT) sensor for automated repositioning of a motorized XY stage to keep a moving fluorescent marker signal in the field of view (*Faumont et al., 2011*). For this purpose, an oblique 80% transmission filter was inserted in the light path to divert 20% of the light to the PMT quadrants. A 535/30 bandpass filter (F47-535, AHF) was used to narrow the emission spectrum prior to detection for improved tracking performance. A fluorescent marker GFP was expressed in vulval muscle cells, strains PD4665 and ZX2012. Video files containing data of both fluorescent channels (for GCaMP6 and mCherry) were processed with custom written Wolfram Mathematica notebooks. Both color channels were virtually overlaid to accurately correct the spatial alignment. Images were first binarized to identify the centroid of the moving neuronal cell bodies throughout all frames. Mean intensity values of a circular ROI (18 pixel radius) centered on this centroid were measured and subtracted with the mean intensity values of a surrounding donut shaped background ROI (five pixel width). Coordinates of two AS neurons of interest were recorded relative to the vulva and to each other to obtain their relative distance and the angle between the vulva and the two neurons of interest. The traces were normalized to $\Delta F/F = (F_i- F)/F$, where F represents the average of the entire trace, and were used for correlation analysis. We used strain ZX2435, raised on ATR, to initiate long reversals. This strain expresses the step-function opsin ChR2(C128S) in the backward PINs including AVA (*Schultheis et al., 2011*).

## Measurement of Ca$^{2+}$ signals in PINs

Ca$^{2+}$ imaging with cameleon D3cpv (*Palmer et al., 2006*) was performed on 5% agar pads as described (*Kawano et al., 2011*) on an Axiovert 200 microscope (Zeiss), using a 100x/1.30 EC Plan-Neofluar Oil M27 oil immersion objective. ChR2 stimulation was performed using 8 mW/mm$^2$ blue light delivered by a 100 W HBO mercury lamp. The excitation light path was split using a dual-view (Photometrics) beam splitter with a CFP/YFP filter set. The YFP/CFP ratio after background subtraction was normalized to the $\Delta R/R=(R_i-R)/R$, where $R_i$ represents the YFP/CFP ratio at the given time point and R represents the average of the entire trace during blue light stimulation. YFP/CFP ratios without normalization were used for quantification and statistics (*Figure 6BIV*). This data did not pass the Shapiro-Wilk normality test.

## Correlation analysis

Cross-correlation analyses were performed with built-in MATLAB functions. $Ca^{2+}$ transients in AS6 and AS7 and vulva angles were smoothed for 10 frames. Individual bending events identified as segment of the trace between two minima were used for cross-correlation with 100 time lags (10 s). For comparison of peak correlations, the maximum correlation (positive or negative) in a 5 s time window centered on the peak of the control mean correlation was used.

## Statistics

Data is given as means ± SEM. Significance between data sets after two-tailed Student's t-test or after Mann-Whitney U-test or after ANOVA is given as p-value (*$p \leq 0.05$; **$p \leq 0.01$; ***$p \leq 0.001$), the latter was given following Tukey's post-hoc test. Data was analyzed and plotted in Excel (Microsoft, USA), in OriginPro 2016 (OriginLab Corporation, Northampton, USA) or in MATLAB (MathWorks, Natick, Massachusetts, USA).

## Acknowledgements

We thank Cori Bargmann for suggesting to study the AS MNs. We are grateful to Mei Zhen, Kang Shen, Xing Wei, Navin Pokala, Cori Bargmann, Yishi Jin, and Andrew Chisholm for reagents and to Isabell Franz, Mona Hoeret, Heike Fettermann, Regina Wagner and Heinz Schewe for expert technical assistance. Yongmin Cho, Daniel Porto and Hang Lu provided equipment and software. We thank Gal Haspel for fruitful discussions. This work was funded by a GO-IN stipend of Goethe University, in conjunction with the EU program PCOFUND-GA-2011 – 291776, GO-IN (to OT), by a IMPReS PhD stipend (to A.B.) and by grants GO1011/4-2 (Protein-based Photoswitches), GO1011/8-1 (NewOptogeneticsTools) and EXC115/3 (Cluster of Excellence Frankfurt - Macromolecular Complexes) from the Deutsche Forschungsgemeinschaft (DFG) to AG.

## Additional information

### Funding

| Funder | Grant reference number | Author |
| --- | --- | --- |
| Deutsche Forschungsgemeinschaft | GO1011/4-2 | Petrus Van der Auwera Wagner Steuer Costa Alexander Gottschalk |
| Goethe University | GO-IN | Oleg Tolstenkov |
| European Union Marie Curie Actions | PCOFUND-GA-2011-291776 | Oleg Tolstenkov |
| Deutsche Forschungsgemeinschaft | GO1011/8-1 | Oleg Tolstenkov Alexander Gottschalk |
| Deutsche Forschungsgemeinschaft | EXC115/3 | Petrus Van der Auwera Wagner Steuer Costa Alexander Gottschalk |
| Max Planck Research School | IMPReS Membrane Biology | Amelie CF Bergs |

The funders had no role in study design, data collection and interpretation, or the decision to submit the work for publication.

### Author contributions

Oleg Tolstenkov, Conceptualization, Resources, Data curation, Formal analysis, Funding acquisition, Validation, Investigation, Visualization, Methodology, Writing—original draft, Writing—review and editing; Petrus Van der Auwera, Resources, Software, Supervision, Methodology; Wagner Steuer Costa, Software, Formal analysis, Methodology; Olga Bazhanova, Data curation, Formal analysis; Tim M Gemeinhardt, Resources, Data curation, Formal analysis, Methodology; Amelie CF Bergs, Resources; Alexander Gottschalk, Conceptualization, Formal analysis, Supervision, Funding acquisition, Validation, Visualization, Project administration, Writing—review and editing

## Author ORCIDs

Oleg Tolstenkov (ID) http://orcid.org/0000-0002-6484-9965
Petrus Van der Auwera (ID) http://orcid.org/0000-0001-7540-4788
Wagner Steuer Costa (ID) http://orcid.org/0000-0001-7707-2596
Alexander Gottschalk (ID) http://orcid.org/0000-0002-1197-6119

## Decision letter and Author response

Decision letter https://doi.org/10.7554/eLife.34997.036
Author response https://doi.org/10.7554/eLife.34997.037

# Additional files

## Supplementary files

• Transparent reporting form
DOI: https://doi.org/10.7554/eLife.34997.027

## Data availability

All data generated or analysed during this study are included in the manuscript and supporting files. Source data files are videos from live cell imaging and behavioral experiments. Due to their size, they are provided via the following link (sorted by main figure of the paper): https://open.ag.bmls. uni-frankfurt.de/s/x2ixrBokXazi6wQ.

The following dataset was generated:

| Author(s) | Year | Dataset title | Dataset URL | Database, license, and accessibility information |
| --- | --- | --- | --- | --- |
| Oleg Tolstenkov, Petrus Van der Auwera, Wagner Steuer Costa, Olga Bazhanova, Tim M Gemeinhardt, Amelie CF Bergs, Alexander Gottschalk | 2018 | Behavior and imaging videos from which data in this paper was generated, sorted by figure number | https://open.ag.bmls. uni-frankfurt.de/s/x2ixr-BokXazi6wQ | Publicly available at institutional server: open.ag.bmls.uni-frankfurt.de |

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
