## [Decision Letter]

Thank you for submitting your article "Functionally asymmetric motor neurons coordinate locomotion of *Caenorhabditis elegans*" for consideration by *eLife*. Your article has been reviewed by two peer reviewers, and the evaluation has been overseen by Oliver Hobert as the Reviewing Editor and Eve Marder as the Senior Editor. The following individuals involved in review of your submission have agreed to reveal their identity: Mei Zhen (Reviewer #1); Gal Haspel (Reviewer #2).

Both reviewers, as well as the BRE, see great value in the description of the long sought function of the AS motor neurons in *C.elegans* and agree that this manuscript is of potential interest for publication in *eLife*. However, you will see that both reviewers have substantial comments that we ask you to address in detail. Apart from critical controls, it is essential to add discussions to highlight potential caveats and to remove conclusions drawn from somewhat ambiguous experimental results.

Reviewer #1:

Tolstenkov et al. reported the functional characterization of the AS class excitatory motor neurons in *C. elegans*. They employed an array of optogenetic tools to assess the effect of perturbation of AS motor neuron activity. Specifically, they demonstrated that optogenetic manipulation of these neurons – either activation or inhibition – led to changes in *C. elegans*' motor pattern. They used calcium imaging to show that the AS motor neurons exhibited cyclic activity during movements, and likely activate sequentially during forward movement. Lastly, combining calcium imaging and optogenetic manipulation of neurons, they showed that AS neurons' anatomic connections with the premotor interneurons, previously described morphologically by the EM reconstruction, could be functional.

As the first report on the property and function of the AS motor neurons, this study begins to fill a significant knowledge gap in the current *C. elegans* motor circuit field.

However, there are several concerns on interpretation of several results that are key to this study's conclusions. I would be glad to, however, hear authors' responses and discuss possible ways to address these concerns. Given the body of work, it is conceivable that the authors stick to the solid conclusions on AS motor neuron function (e.g. activated during forward movements), and their functional connectivity with both forward and reversal premotor interneurons (e.g. AS exhibited calcium change upon AVA or AVB activation), and address/discuss the caveats on their activity during reversals and whether AS neurons are necessary for movement of forward or backward locomotion (see below for rationales).

Below are the main concerns and my suggestions, each accompanied by a brief description of experiments and interpretations by authors.

1) The efficacy of two genetic strategies to achieve AS motor neuron-specific activity manipulation (subsection “Selective expression and activation of optogenetic tools in AS MNs”, Figure 1).

The functional exploration of AS motor neurons has long been hindered by the lack of promoters that allow AS-specific manipulation. These authors have gone to great length to design two strategies to achieve this goal.

Both approaches depended on 'subtraction': authors drove the expression of opotogenetic reagents (ChR2 for activation, ACR1/HisCl channel for inhibition, and miniSOG for ablation) in all cholinergic neurons by Punc-17, followed by either reducing the gene expression in non-AS motor neurons by RNAi, or suppressing the gene expression in non-AS motor neurons by the Q system. By restricting light stimulation to the body, where AS neurons reside, they further reduced the chance of perturbation of cholinergic neurons whose soma reside outside the VNC.

The functional assessment of AS motor neuron using these strategies thus depends on the efficient suppression of expression of optogenetic tools in other cholinergic motor neurons, hence the specificity and strength of Pacr-5 and Punc-4 promoters. Our Pacr-5 promoter is relatively weaker than Punc-4. My more critical concern is for Punc-4's effectiveness in adults. This promoter was used to suppress the expression in the A class motor neurons, which share the most extensive overlap in the wiring input and output with AS neurons. The Punc-4 that we obtained from the published sources exhibits robust expression in A motor neurons until the animals reached the L4 stage. As animals enter the adult stage, Punc-4 becomes drastically reduced in A motor neurons, and robustly turned on in VCs. If the authors conducted all experiments (activation, inhibition, and ablation) in animals after the L4 stage, there could be insufficient depletion of gene expression to allow residual functional contribution of the A motor neurons (in the case of ChR2 and ACR1/histamine-Cl manipulation), and of the A motor neuron functional loss (in the case of miniSOG induced neuronal ablation), to the observed changes in motor phenotypes.

Can the authors describe details on their strategies that ensure the specificity and efficacy of a depletion of gene expression in the A and B class motor neurons, and that the behavioral effects in these experiments could be confidently attributed to the effect of manipulation of the AS motor neurons alone?

If proper controls for the expression level turn out to be too difficult to assess, or informative, a potential circumvent may be to compare the effect on the locomotion of L4 larvae when either activating or inhibiting AS neurons using the Q system. This experiment is fairly feasible, and the outcome could be fairly informative (this suggestion also applies to point #4).

2) The change in locomotion upon inhibition of AS motor neuron function (subsection “AS MN ablation disrupts the locomotion pattern”; Figure 3).

A notion that authors did not discuss is the striking difference in the severity of effect on locomotion upon the activation and inaction of AS neurons.

ChR2 induced activation of AS neurons did not stop the prevent bend propagation (it seems that manipulations appeared to be applied predominantly when animals were executing the forward movement), but induced a dorsal bias in steering during forward motions. This is in line with the dorsal-muscle-specific output of the AS neurons as predicted from the wiring diagram, as well as the reported results of VD ablation, which leads to ventral biased steering.

However, the disruption of the AS motor neuron function – by miniSOG, ACR1, or histamine – all led to severe stalling of bending wave propagation. To me, this is a bit surprising. If these effects were solely due to a perturbation of the AS neurons, then AS neurons play a major role in driving sequence body bending during forward locomotion. To me, this does not easily reconcile with the mild effect induced by AS activation.

A major difference is the sensitivity of ontogenetic tools. ChR2 has a marked reduced efficacy than chrimson for evoking behavioral responses. Like chrimson, the sensitivity of ACRs is raising our vigilance on the expression specificity and strength promoters used to drive the optogenetic tool's expression. For both chrimson and ACRs, we found they could induce robust behavioral changes from neurons that we could barely detect expression with fluorescent tags, unless we perform antibody staining. With these reagents, the 'contamination' effect from other cholinergic motor neurons due to insufficient inhibition would be more profound.

3) Calcium recording of AS motor neurons in moving animals (subsection “Oscillatory AS MN activity correlates more strongly with body bends during forward crawling”, Figure 5).

By expressing a genetic calcium sensor in AS motor neurons, authors observed cyclic calcium changes in the soma of AS neurons. By examining the phasic relationship between anteriorly and posteriorly situated AS neurons, authors reported a correlated anterior to posterior activation order of AS neurons during the forward movement, but found no clear phasic relations during reversals. This led to the authors' conclusion that AS neurons sequentially activate, from anterior to posterior during forward movement, but do not follow a reversed sequential order during reversals. Together with the results of AVA and AVB activation (see point 4), these results seem to implicate a more important role of driving forward locomotion by AS motor neurons.

Cautions should be applied to the analyses and interpretation of these calcium recordings. The AS calcium signal changes during reversals, at least for the sample trace, was shown in a short event between two long forward movements. This is a typical profile that I would call the 'transitions'.

The standard correlation analyses applied in this experiments were more suitable for the forward periods, which typically were long events with multiple complete body bend propagation. But we have found it not so for analyzing short periods of transitions. In this sample trace, both AS neurons were changing their phases. The significant delay of soma calcium signal change during an event when the animal switched from forward to reversal, and back to forward, without completing at least two fully reversed body bend propagation prevented observing a clear trend in their phasic relationship for reversals.

To draw more meaningful comparison on AS motor neuron's activity during forward and backward movements, it requires analyses of their activity profiles when animals exhibited longer periods of reversal movements, as in the case for forward movements, where multiple fully propagated head to tail (and vice versa) bending waves were executed. These events were rare and difficult to obtain.

In my opinion, the authors could simply stick to the conclusion that AS neurons are active during forward movements, and exhibited potential sequential activation that correlates with anterior to posterior body bend propagation.

4) Reciprocal and asymmetric activation of AVA, AVB, and AS (subsection “AS MNs integrate signaling from both forward and backward premotor-interneurons”, Figure 6).

In this set of experiments, the authors applied ChR2 activation of AVA, or of AVB (with other neurons by Psra-11), while recording the AS motor neuron activity by calcium imaging. They observed a strong increase of calcium signals in AS neurons by Psra-11 activation, but only weak increase with AVA activation.

They performed the reciprocal experiments. When they activated AS motor neurons, they observed a strong increase of calcium signals in AVA, but only a weak increase in AVB. From these results, the authors concluded asymmetry – where AVB dominants activation to AS, and AS retrogradely activates predominantly AVA.

These results are not in immediate consensus with the reported wiring diagram, where AS receive predominantly synaptic inputs, with large numbers of gap junctions and chemical synapses from the reversal premotor interneurons (AVA and AVE), whereas the inputs from the forward premotor interneurons are small (5 gap junctions with AVB and 2 chemical synapses from PCV (subsection “AS MNs integrate signaling from both forward and backward premotor-interneurons”).

The merit of these experiments was the functional demonstration of the reported anatomic connectivity, which I strongly support. But I would be very cautious about extrapolating any functional relationship among these neurons by comparing the extent of calcium changes observed from these experiments. They key caveat that could profoundly affect the outcome of the calcium recording is the different expression levels of both ChR2 and the calcium sensors.

In the case of AVA versus AVB ChR2 stimulation, one could not compare their effect on AS activation without calibrating the effect of ChR2 on AVA and AVB first. Without knowing the difference in their physiological properties, and with ChR2 driven by two different promoters, this comes close to being impossible to calibrate. A potential approximation, or crude control, would be to calibrate the light intensity used for activation to alter behavior, specifically, at which intensity to AVA-CHR2 or AVB-CHR2 could reliably induce reversal or forward. From my understanding, the AVA specific ChR2 line (as reported in a previous paper cited by the authors) is fairly weak in expression. A lesser effect on inducing AS calcium signal change may be simply attributed to a less activated AVA, than AVB in these experiments.

A similar concern resides with comparing the calcium signal changes in AVA and AVB by AS activation. In this case, both reporters were cameleon (Kawano et al. 2011), which were driven by promoters with drastically different expression levels for AVA and AVB than above. The Pnmr-1 promoter used for AVA recording in this strain exhibited strong expression in AVA and AVE, but the Psra-11 promoter used for AVB recording here exhibited much weaker expression in AVB (the strong expression is in another neuron AIY). Using these two strains, AVA activity exhibited changes that were fast and large when animals transit to reversals, but AVB activity changes were slow and small during transition from reversal to forward (Kawano et al. 2011). Note that during the AVA and AVB simultaneous co-imaging, two different change scales were used to report calcium changes for AVA and AVB (Kawano et al. 2011). The large and small responses in AVA and AVB upon AS stimulation may simply reflect the nature of these reporters.

The observation – activation of one neuron leads to calcium signal changes in another – itself is a strong evidence that the anatomic connection between the two neurons are functional. But comparing the effect of stimulation of two different neurons, without careful calibration of reagents, could lead to wrong conclusions. Authors should stick simply to the observation that activation of both AVA and likely AVB could lead to activation of AS neurons, and optogenetic depolarization of AS was able to induce calcium changes in AVA, supporting reciprocal activation between AVA and AS.

Reviewer #2:

The authors shed light on the function of the often neglected AS motoneuron class of *C. elegans*. Historically, this motoneuron class has been kept out of many simplified diagrams, models, and studies. It is an important contribution that will hopefully allow researchers to include AS in their models and hypotheses. The narrative is clear and experiments are well designed to use complimentary techniques.

My main concerns are the emphasis on CPG, that although dear to my heart, were not found or specifically looked for in this study, and the possible bias of conclusions about backward locomotion emerging from the fact that most data collected is from forward locomotion.

1) The principal function of the forward locomotion circuit is presented in the Introduction and Discussion as if it has been concluded that a head oscillator drives sensory mediated propagation. That idea was suggested (as reviewed by us and others) but not well supported yet.

2) Electrophysiology of muscle during AS activation is important and quantitative data should be presented. Ideally, current parameters should be calculated and compared to existing data from the authors' studies when all cholinergic MN were activated.

3) On the other hand, the authors conclusion that AS are not pacemaker because "did not induce any obvious intrinsic rhythmic activity" is premature and should be omitted for three reasons: the scarcity of presented data, the possibility that conditions are not appropriate for oscillations to occur (neuromodulation, descending input), and that other excitatory MN that are now thought to produce oscillations fail at this specific test.

4) Optogenetic activation of AS seems to have more complex effect on muscle activity than presented in the text. The effect on muscle is transient and a ventral activation (maybe alternating) seems to begin after 2.5 seconds.

5) MiniSOG results are not presented in the same format as optogenetic activation or histamine inactivation. Was Ca^2+^ imaging done in the miniSOG strain? If not possible, mention the reason.

6) The authors seem to suggest that disinhibition of ventral muscle induced calcium fluctuations of large amplitude (subsection “Chronic hyperpolarization of AS MNs eliminates Ca^2+^ activity in dorsal BWMs”). If fluctuations are rhythmic, this is a major finding because it suggests pacemaker activity (of non-AS motoneurons) that is uncoupled from the alternating antagonistic.

7) Some conclusions are drawn regarding forward and backward behavior while most data collected has been for forward. A reasonable solution for Figure 5 (other than inducing long bouts of backward locomotion with an irritant such as quinine or specific feeding state) is to randomly pick the same number of locomotion cycles from forward to statistically compare to the backward data.

8) Conclusions from optogenetic stimulation of PIN, comparing effects of AVA vs. AVB stimulation should consider that the experiment was performed in two different transgenic strains and that other PINs in the forward and backward groups were not activated. Also, similar to the retrograde experiment, can gj mutant be used to see the contribution of chemical and electrical signals?

---

## [Author Response]

Reviewer #1:[…] Below are the main concerns and my suggestions, each accompanied by a brief description of experiments and interpretations by authors.1) The efficacy of two genetic strategies to achieve AS motor neuron-specific activity manipulation (subsection “Selective expression and activation of optogenetic tools in AS MNs”, Figure 1).The functional exploration of AS motor neurons has long been hindered by the lack of promoters that allow AS-specific manipulation. These authors have gone to great length to design two strategies to achieve this goal.Both approaches depended on 'subtraction': authors drove the expression of opotogenetic reagents (ChR2 for activation, ACR1/HisCl channel for inhibition, and miniSOG for ablation) in all cholinergic neurons by Punc-17, followed by either reducing the gene expression in non-AS motor neurons by RNAi, or suppressing the gene expression in non-AS motor neurons by the Q system. By restricting light stimulation to the body, where AS neurons reside, they further reduced the chance of perturbation of cholinergic neurons whose soma reside outside the VNC.The functional assessment of AS motor neuron using these strategies thus depends on the efficient suppression of expression of optogenetic tools in other cholinergic motor neurons, hence the specificity and strength of Pacr-5 and Punc-4 promoters. Our Pacr-5 promoter is relatively weaker than Punc-4. My more critical concern is for Punc-4's effectiveness in adults. This promoter was used to suppress the expression in the A class motor neurons, which share the most extensive overlap in the wiring input and output with AS neurons. The Punc-4 that we obtained from the published sources exhibits robust expression in A motor neurons until the animals reached the L4 stage. As animals enter the adult stage, Punc-4 becomes drastically reduced in A motor neurons, and robustly turned on in VCs. If the authors conducted all experiments (activation, inhibition, and ablation) in animals after the L4 stage, there could be insufficient depletion of gene expression to allow residual functional contribution of the A motor neurons (in the case of ChR2 and ACR1/histamine-Cl manipulation), and of the A motor neuron functional loss (in the case of miniSOG induced neuronal ablation), to the observed changes in motor phenotypes.

For RNAi, this is probably a valid concern, though it is often claimed that RNAi is a self-amplifying mechanism, so it might be sustained even after the *unc-4* promoter is turned off. For the Q system, it should depend on how persistent the QS transcriptional suppressor protein is. If it is sufficiently stable, after generation in/before L4 stage, it may be present in sufficient concentration in adults.

Can the authors describe details on their strategies that ensure the specificity and efficacy of a depletion of gene expression in the A and B class motor neurons, and that the behavioral effects in these experiments could be confidently attributed to the effect of manipulation of the AS motor neurons alone?

First, worms used for experiments were selected by fluorescence microscopy, to ensure downregulation in most cholinergic neurons. Of course, efficacy of downregulation is difficult to be verified based on a fast inspection of fluorescence. However, we have exemplary analyzed the expression pattern of ChR2 in AS motor neurons in more detail, following either the RNAi approach or the Q system approach. The data is now included as a new supplementary figure (Figure 1—figure supplement 1). Since it is not straightforward to identify AS neurons in each individual that is analyzed in the behavioral experiments, we performed this analysis on a number of individuals, and show the result in a gallery. Also, we counted the cell bodies that were readily visible based on ChR2::YFP fluorescence. In both the RNAi animals as well as the Q system animals, the number of cell bodies visible in the VNC was reduced by 2/3, i.e. from ca. 40 in the parental transgene expressing animals to 12 and 10 in the animals expressing RNAi constructs for ChR2, or the Q System approach.

If proper controls for the expression level turn out to be too difficult to assess, or informative, a potential circumvent may be to compare the effect on the locomotion of L4 larvae when either activating or inhibiting AS neurons using the Q system. This experiment is fairly feasible, and the outcome could be fairly informative (this suggestion also applies to point #4).

We now performed several experiments in L4 larval stage (as suggested by reviewer 1) and provide more detailed analyses of expression patterns in the manuscript (Figure 1—figure supplement 1). We compared several experiments in L4 stage to the ones we performed in adults, e.g. with ChR2 and ACR1 in AS neurons (behavior) and with ChR2 in AS neurons (Ca^2+^ imaging in AVA and AVB). Here we got essentially the same results as in wt, and no statistically significant differences were observed. In addition, we performed experiments with Chrimson in AS neurons, which we included as a new supplementary figure in the manuscript (Figure 1—figure supplement 3). For Chrimson, we see overall stronger effects, however, no gross alterations in the qualitative results or in altering the propagation of the body wave. However, we observed an imminent problem with using Chrimson in that it is so light-sensitive: It was difficult to prevent the animals being pre-activated by background light, in particular, as we use an illumination device based on a LCD projector and not a DMD. This appears to partially activate Chrimson (we see differences in baseline speed upon addition of ATR), and may activate Chrimson also in those cells that we want to keep inactive, i.e. those in head and tail ganglia.

**Author response image 1. respfig1:** ChR2 expressed and specifically photoactivated in AS neurons pf L4 larvae. Light-evoked effects on body length, locomotion speed and body bending angles.

**Author response image 2. respfig2:** ACR1 expressed and specifically photoactivated in AS neurons of L4 larvae. Light-evoked effects on body length and locomotion speed.

**Author response image 3. respfig3:** ChR2 and ACR1 expressed and specifically photoactivated in AS neurons in L4 larvae. Light evoked effects on body length and locomotion speed. Comparisons to the same experiments performed in adult animals, as reported in the manuscript.

**Author response image 4. respfig4:** ChR2 expressed and specifically photoactivated in AS neurons of L4 larvae, imaging of Ca^2+^ transients in AVA or AVB neurons, using Cameleon. Compare to Figure 6C in the manuscript (formerly 6B, data for adult animals).

2) The change in locomotion upon inhibition of AS motor neuron function (subsection “AS MN ablation disrupts the locomotion pattern”; Figure 3).A notion that authors did not discuss is the striking difference in the severity of effect on locomotion upon the activation and inaction of AS neurons.ChR2 induced activation of AS neurons did not stop the prevent bend propagation (it seems that manipulations appeared to be applied predominantly when animals were executing the forward movement), but induced a dorsal bias in steering during forward motions. This is in line with the dorsal-muscle-specific output of the AS neurons as predicted from the wiring diagram, as well as the reported results of VD ablation, which leads to ventral biased steering.However, the disruption of the AS motor neuron function – by miniSOG, ACR1, or histamine – all led to severe stalling of bending wave propagation. To me, this is a bit surprising. If these effects were solely due to a perturbation of the AS neurons, then AS neurons play a major role in driving sequence body bending during forward locomotion. To me, this does not easily reconcile with the mild effect induced by AS activation.A major difference is the sensitivity of ontogenetic tools. ChR2 has a marked reduced efficacy than chrimson for evoking behavioral responses. Like chrimson, the sensitivity of ACRs is raising our vigilance on the expression specificity and strength promoters used to drive the optogenetic tool's expression. For both chrimson and ACRs, we found they could induce robust behavioral changes from neurons that we could barely detect expression with fluorescent tags, unless we perform antibody staining. With these reagents, the 'contamination' effect from other cholinergic motor neurons due to insufficient inhibition would be more profound.

This is a valid concern. ACR1 could induce effects even from spurious expression. However, the fact that acute hyperpolarization of AS MNs by ACR1 leads to contraction, which can be reverted to elongation in the *unc-47* mutant, speaks against effects originating from ‘contaminating’ expression, as these effects should have not been affected that much by the *unc-47* mutation.However, we did perform experiments with Chrimson (new Figure 1—figure supplement 3). As stated above, this protein is highly light sensitive and mediates stronger currents when compared to ChR2. We could recognize this because the basal crawling speed of animals (raised with ATR) was reduced when we placed them on our single-worm tracking and illumination system (Stirman et al., 2011). This system uses a LCD projector, and not a digital mirror device (DMD) to produce light patterns, and thus some low amounts of light may reach the animal before stimulation. For ChR2, this is not a problem, as the light intensities are too low. This, however, emphasizes that stronger activation of AS neurons should be expected, which is what we observed from our experiments (see Figure 1—figure supplement 3). Bending angles were increased, as well as body contraction, and speed was more reduced compared to animals expressing ChR2 in the AS neurons (while the effect was still only transient), when illumination was turned on. However, compared to AS::ChR2 expressing animals, the AS::Chrimson animals were still able to move essentially normally (see new Video 2; other video numbers went up by one) and propagated the body wave while Chrimson was photostimulated. This argues that even stronger photostimulation of AS neurons does not disrupt the bending wave. The fact that inhibition or ablation of the AS neurons has much stronger effects implies that hyperpolarization may affect other cells through gap junctions. Possibly, (some of) the junctions are rectifying towards AS neurons, thus these may act as a current sink, while they may not allow much current to flow from AS neurons out, and that disturbance of this electrical network when AS neurons are missing is having strong effect on the connected cells. These effects may add up with effects of missing activation of dorsal muscle and VD GABAergic neurons, when AS neurons are inactivated. In sum, inhibition appears to evoke stronger effects on the body wave than (even stronger) stimulation of the AS neurons.

3) Calcium recording of AS motor neurons in moving animals (subsection “Oscillatory AS MN activity correlates more strongly with body bends during forward crawling”, Figure 5).By expressing a genetic calcium sensor in AS motor neurons, authors observed cyclic calcium changes in the soma of AS neurons. By examining the phasic relationship between anteriorly and posteriorly situated AS neurons, authors reported a correlated anterior to posterior activation order of AS neurons during the forward movement, but found no clear phasic relations during reversals. This led to the authors' conclusion that AS neurons sequentially activate, from anterior to posterior during forward movement, but do not follow a reversed sequential order during reversals. Together with the results of AVA and AVB activation (see point 4), these results seem to implicate a more important role of driving forward locomotion by AS motor neurons.Cautions should be applied to the analyses and interpretation of these calcium recordings. The AS calcium signal changes during reversals, at least for the sample trace, was shown in a short event between two long forward movements. This is a typical profile that I would call the 'transitions'.The standard correlation analyses applied in this experiments were more suitable for the forward periods, which typically were long events with multiple complete body bend propagation. But we have found it not so for analyzing short periods of transitions. In this sample trace, both AS neurons were changing their phases. The significant delay of soma calcium signal change during an event when the animal switched from forward to reversal, and back to forward, without completing at least two fully reversed body bend propagation prevented observing a clear trend in their phasic relationship for reversals.To draw more meaningful comparison on AS motor neuron's activity during forward and backward movements, it requires analyses of their activity profiles when animals exhibited longer periods of reversal movements, as in the case for forward movements, where multiple fully propagated head to tail (and vice versa) bending waves were executed. These events were rare and difficult to obtain.

We now acquired more imaging data, and accumulated more reversal events, that lasted longer (>10 s). An example (a 20 s reversal is now shown in the updated Figure 6C). The analysis of longer reversal events indeed altered the outcome, and it appears as if AS neuron function is also correlated with reversal locomotion, albeit the correlation coefficients are lower than for forward locomotion. Yet they are not significantly different to forward. So it appears that AS neurons are involved in both forward and reverse locomotion. We thank the reviewer for suggesting this more extensive analysis.

In my opinion, the authors could simply stick to the conclusion that AS neurons are active during forward movements, and exhibited potential sequential activation that correlates with anterior to posterior body bend propagation.

Based on the new data, it seems as if one may conclude that AS neuron function is correlated with forward as well as reverse locomotion.

4) Reciprocal and asymmetric activation of AVA, AVB, and AS (subsection “AS MNs integrate signaling from both forward and backward premotor-interneurons”, Figure 6).In this set of experiments, the authors applied ChR2 activation of AVA, or of AVB (with other neurons by Psra-11), while recording the AS motor neuron activity by calcium imaging. They observed a strong increase of calcium signals in AS neurons by Psra-11 activation, but only weak increase with AVA activation.They performed the reciprocal experiments. When they activated AS motor neurons, they observed a strong increase of calcium signals in AVA, but only a weak increase in AVB. From these results, the authors concluded asymmetry – where AVB dominants activation to AS, and AS retrogradely activates predominantly AVA.These results are not in immediate consensus with the reported wiring diagram, where AS receive predominantly synaptic inputs, with large numbers of gap junctions and chemical synapses from the reversal premotor interneurons (AVA and AVE), whereas the inputs from the forward premotor interneurons are small (5 gap junctions with AVB and 2 chemical synapses from PCV (subsection “AS MNs integrate signaling from both forward and backward premotor-interneurons”).The merit of these experiments was the functional demonstration of the reported anatomic connectivity, which I strongly support. But I would be very cautious about extrapolating any functional relationship among these neurons by comparing the extent of calcium changes observed from these experiments. They key caveat that could profoundly affect the outcome of the calcium recording is the different expression levels of both ChR2 and the calcium sensors.

We toned down the conclusions for these experiments, along the lines suggested, as we agree with reviewer 1, i.e. that expression levels of the optogenetic actuator in the different cell types (AVA and AVB) may be different and this could affect the outcome of the experiment (subsection “AS MNs integrate signaling from both forward and backward PINs”, second paragraph and subsection “AS MNs act in coordination of dorso-ventral bends, antero-posterior wave propagation and possibly forward-backward states”, last paragraph).

In the case of AVA versus AVB ChR2 stimulation, one could not compare their effect on AS activation without calibrating the effect of ChR2 on AVA and AVB first. Without knowing the difference in their physiological properties, and with ChR2 driven by two different promoters, this comes close to being impossible to calibrate. A potential approximation, or crude control, would be to calibrate the light intensity used for activation to alter behavior, specifically, at which intensity to AVA-CHR2 or AVB-CHR2 could reliably induce reversal or forward. From my understanding, the AVA specific ChR2 line (as reported in a previous paper cited by the authors) is fairly weak in expression. A lesser effect on inducing AS calcium signal change may be simply attributed to a less activated AVA, than AVB in these experiments.

We agree this type of calibration is not really feasible, thus we toned down our conclusions (see above, and subsection “AS MNs integrate signaling from both forward and backward PINs”, second paragraph).

A similar concern resides with comparing the calcium signal changes in AVA and AVB by AS activation. In this case, both reporters were cameleon (Kawano et al. 2011), which were driven by promoters with drastically different expression levels for AVA and AVB than above. The Pnmr-1 promoter used for AVA recording in this strain exhibited strong expression in AVA and AVE, but the Psra-11 promoter used for AVB recording here exhibited much weaker expression in AVB (the strong expression is in another neuron AIY). Using these two strains, AVA activity exhibited changes that were fast and large when animals transit to reversals, but AVB activity changes were slow and small during transition from reversal to forward (Kawano et al. 2011). Note that during the AVA and AVB simultaneous co-imaging, two different change scales were used to report calcium changes for AVA and AVB (Kawano et al. 2011). The large and small responses in AVA and AVB upon AS stimulation may simply reflect the nature of these reporters.

This is of course a valid concern. Possibly this could be due to Ca^2+^ buffering upon higher expression of cameleon – however the signal was weak in AVB, where also the expression was low. As the sensor is ratiometric, the signal should not depend too much on expression levels? Maybe AVB activity is simply different from AVA. Of course, one should not conclude this from the presently available data, so we restricted our conclusion to saying that AVA receives electrical feedback from AS MNs.

One thing we did observe is an apparent signal in AVB during the first few seconds of the experiment. This is likely due to fast bleaching of a fraction of the CFP molecules. Normally, in Ca^2+^ imaging experiments, one typically waits for a stable baseline signal, before beginning to measure or to apply a stimulus. However, here, as the imaging light also sets off ChR2 activation in AS MNs, we need to measure immediately. Importantly, the apparent signal is also seen without ATR, so it is not AS MN dependent. We saw it also in L4 animals (see Author response image 4). One can correct the data, as shown in Author response image 5, but as we reported the non-corrected data earlier, we did not want to alter this now.

**Author response image 5. respfig5:** Figure R5. ChR2 expressed and specifically photoactivated in AS neurons, imaging of Ca^2+^ transients in AVB neurons of L4 larvae, using Cameleon. Apparent photobleaching of CFP during the first 1-2 seconds was corrected for.

The observation – activation of one neuron leads to calcium signal changes in another – itself is a strong evidence that the anatomic connection between the two neurons are functional. But comparing the effect of stimulation of two different neurons, without careful calibration of reagents, could lead to wrong conclusions. Authors should stick simply to the observation that activation of both AVA and likely AVB could lead to activation of AS neurons, and optogenetic depolarization of AS was able to induce calcium changes in AVA, supporting reciprocal activation between AVA and AS.

We followed these suggestions of reviewer 1. Thank you.

Reviewer #2:The authors shed light on the function of the often neglected AS motoneuron class of C. elegans. Historically, this motoneuron class has been kept out of many simplified diagrams, models, and studies. It is an important contribution that will hopefully allow researchers to include AS in their models and hypotheses. The narrative is clear and experiments are well designed to use complimentary techniques.My main concerns are the emphasis on CPG, that although dear to my heart, were not found or specifically looked for in this study, and the possible bias of conclusions about backward locomotion emerging from the fact that most data collected is from forward locomotion.1) The principal function of the forward locomotion circuit is presented in the Introduction and Discussion as if it has been concluded that a head oscillator drives sensory mediated propagation. That idea was suggested (as reviewed by us and others) but not well supported yet.

We agree, we are also not aware of rock-solid evidence of a head CPG, though the RIA and SMB/SMD neurons, as shown by Yun Zhang’s laboratory, have oscillatory activity. Yet, this was in animals with unrestrained head, that exhibit head oscillations, so one would not want to conclude this to be intrinsic activity. We thus toned down the text in Introduction (third paragraph) and Discussion (subsection “AS MNs as part of CPGs in the ventral nerve cord?”). In a recent presentation from the Zimmer lab at the worm neurobiology meeting, oscillatory activity in head motor neurons was shown in immobilized animals. Yet, as this is on timescales slower than the actual movement would be in a free-moving animal and as this work is not yet published, we refrain from / cannot refer to this finding.

2) Electrophysiology of muscle during AS activation is important and quantitative data should be presented. Ideally, current parameters should be calculated and compared to existing data from the authors' studies when all cholinergic MN were activated.

We have recorded > 50 animals, in the way described (i.e. patching dorsal muscle, cutting ventrally on the left side of the body, trying to keep the circuits and innervation to muscles as intact as possible), but received only this one measurement showing activation by ChR2. This is why we did not perform any statistical analysis (even though we could determine mean amplitude and rate of minis also from this measurement, it is only one animal). Of course, this is not very robust, yet given the many recordings we have obtained in earlier work after photostimulation of the *zxIs6* transgene, we were quite convinced that this is the activity we could expect from a cholinergic neuron innervating muscle. Nevertheless, we now removed this measurement from the manuscript, rather than keeping it in there and describing the fact that this is only a single measurement (though we are open for any suggestions by the reviewers or the editor). Consequently, Dr. Liewald is no longer an author.

3) On the other hand, the authors conclusion that AS are not pacemaker because "did not induce any obvious intrinsic rhythmic activity" is premature and should be omitted for three reasons: the scarcity of presented data, the possibility that conditions are not appropriate for oscillations to occur (neuromodulation, descending input), and that other excitatory MN that are now thought to produce oscillations fail at this specific test.

We removed this notion.

4) Optogenetic activation of AS seems to have more complex effect on muscle activity than presented in the text. The effect on muscle is transient and a ventral activation (maybe alternating) seems to begin after 2.5 seconds.

The data we show in Figure 2BIII (previously 2AIII) may indicate that some slight oscillations follow after the initial activation of AS neurons. These analyses extended to longer time scales, providing a hint of oscillations, however, when we chose to take the average of the entire trace for normalization of the data, and not only the time at the start (where AS depolarization started), the apparent signals were within the range of the noise, thus we cannot conclude much more from our analysis (but see below).

5) MiniSOG results are not presented in the same format as optogenetic activation or histamine inactivation. Was Ca^2+^ imaging done in the miniSOG strain? If not possible, mention the reason.

We now included muscle Ca^2+^ imaging data for AS::miniSOG ablated animals, see Figure 3A VI-VII. Significantly different mean Ca^2+^ signals were observed between ventral and dorsal muscle, upon AS neuron ablation, reflecting the missing drive to dorsal muscle (Ca^2+^ signals in dorsal were lower, while they were unaffected in ventral muscle).

6) The authors seem to suggest that disinhibition of ventral muscle induced calcium fluctuations of large amplitude (subsection “Chronic hyperpolarization of AS MNs eliminates Ca^2+^ activity in dorsal BWMs”). If fluctuations are rhythmic, this is a major finding because it suggests pacemaker activity (of non-AS motoneurons) that is uncoupled from the alternating antagonistic.

We regularly see oscillating activity in immobilized animals in the muscle ensemble. We have now included example traces in the new version of Figure 2, panels BIII and BVII (these examples are from non-stimulated animals). We followed a suggestion by Dr. Haspel and aligned this activity based on the dorsal Ca^2+^ peaks and indeed see a reciprocal activity on the ventral side (Figure 2BVIII). This is most certainly evoked by motor neurons. We have analyzed this in the context of another project, in which we study effects of a peptidergic neuron on the motor system. We observe Ca^2+^ oscillations in both muscle and cholinergic motor neurons, thus suggesting that neurons are the source of these muscular oscillations, and both types of activity are suppressed upon photoactivation of the peptidergic neuron. We will hopefully be able to present this soon to the general public.

7) Some conclusions are drawn regarding forward and backward behavior while most data collected has been for forward. A reasonable solution for Figure 5 (other than inducing long bouts of backward locomotion with an irritant such as quinine or specific feeding state) is to randomly pick the same number of locomotion cycles from forward to statistically compare to the backward data.

We now performed more experiments and acquired a number of long reversal events (presented in Figure 5C, E-G). See above our response to reviewer 1, who brought up the same point.

8) Conclusions from optogenetic stimulation of PIN, comparing effects of AVA vs. AVB stimulation should consider that the experiment was performed in two different transgenic strains and that other PINs in the forward and backward groups were not activated. Also, similar to the retrograde experiment, can gj mutant be used to see the contribution of chemical and electrical signals?

We now included experiment on AS MN stimulation and recording AVA and AVB neurons in an *unc-7* mutant background (new Figure 6B). This background eliminated significant differences in AS Ca^2+^ signals in stimulated vs. non-stimulated animals, indicating that gap junctions may be significantly contributing to signaling from AVA to AS MNs. This was not the case for AVB, here the *unc-7* mutation had no effect on signals in AS. Nevertheless, given the comments by reviewer 1 on the difficulties in quantitative comparisons due to potentially different levels of transgene expression in AVA and AVB, we toned down conclusions in this section of the manuscript.